# Identifying Synoptic Controls on Boundary Layer Thermodynamic and Cloud Properties in a Regional Forecast Model

Jordan M. Eissner[1], David B. Mechem[1], Yi Jin[2], Virendra P. Ghate[3], and James F. Booth[4]

[1]Department of Geography and Atmospheric Sciences, University of Kansas, Lawrence, KS, 66049, USA
[2]Naval Research Laboratory, Monterey, CA, 93943, USA
[3]Argonne National Laboratory, Lemont, IL, 60439, USA
[4]City University of New York, City College and the Graduate Center, New York, NY, 10031, USA

**Correspondence:** Jordan M. Eissner (jeissner@ku.edu)

**Abstract.** Although most of our understanding of boundary-layer cloudiness is based on idealized, subtropical, barotropic marine environments, boundary-layer clouds exist across a range of conditions. In this study, we use the Naval Research Laboratory's Coupled Ocean/Atmosphere Mesoscale Prediction System (COAMPS) and an automated cold-front-relative analysis framework to explore the boundary layer structure associated with low clouds across a transect through the cold front of a midlatitude synoptic cyclone. The model credibly captures boundary-layer structure in line with conceptual models of midlatitude cyclones and ground-based observations at Graciosa Island in the Azores. The warm sector is conditionally unstable, with clouds that are too shallow and with too little liquid water, compared to cloud-property retrievals from satellite and surface-based instruments. The cold frontal region exhibits convection associated with weak stability and ascent. Northwest of the cold front, the boundary layer is well-mixed, deeper, and capped by a strong inversion maintained by large-scale subsidence. Simulated clouds in frontal and post-frontal regions are mostly too thick, with too much liquid water and too little cloud-base drizzle. The post-frontal clouds are associated with grid-scale updrafts, which appear to be the model's attempt to represent mesoscale organization of cellular convection typically observed in the cold sector of midlatitude cyclones. The deep, well-mixed post-frontal boundary layers and cloudiness are maintained by strong surface fluxes, as in cold air outbreaks. Our analysis framework serves as a unique approach to model verification, and our results offer insights into differences in boundary-layer cloud behavior between subtropical and synoptic cold-sector regimes.

## 1 Introduction

Boundary layer clouds are large contributors to the global energy budget because of their extensive areal coverage and high albedo (Paltridge, 1974; Platt, 1976). A large amount of the cloud feedback uncertainty in global circulation models is due to the misrepresentation of marine low clouds over maritime oceans (Bony and Dufresne, 2005; Zelinka et al., 2016, 2020), including that associated with midlatitude cyclones (Bodas-Salcedo et al., 2012). Much of our understanding of low cloud processes among different cloud regimes has come from observational (Nicholls, 1984; Nicholls and Leighton, 1986; Albrecht et al., 1988; Rauber et al., 2007) and idealized modeling studies (Stevens et al., 2001; Siebesma et al., 2003; Stevens et al., 2005; vanZanten et al., 2011) over barotropic, subtropical regions. However, shallow clouds frequently occur in the cold

sectors of midlatitude cyclones (Field and Wood, 2007; Mechem et al., 2010; Wood, 2012), where the environment is much more spatially inhomogeneous and transient (Naud et al., 2018b), yet these boundary-layer clouds have been much less studied.

In subtropical ocean basins, planetary boundary layer (PBL) and cloud properties are inherently tied to the synoptic environment (Bretherton and Wyant, 1997; Norris and Klein, 2000; Wood and Bretherton, 2006). Conditions in the eastern regions of the semi-permanent high-pressure systems over the subtropical oceans are conducive to the formation of stratocumulus: large-scale subsidence, shallow PBLs, cool sea surface temperatures (SSTs), and weak surface fluxes (Norris and Klein, 2000). Equatorward and westward of the high-pressure center, the stratocumulus clouds transition to shallow cumulus convection. These shallow cumulus have a lower cloud fraction and therefore lower albedo compared to stratocumulus and are associated with warmer underlying SST, stronger surface fluxes, and deeper, decoupled PBLs (Bretherton and Wyant, 1997). Two main mechanisms are invoked to explain this transition. Bretherton and Wyant (1997) noted the importance of latent heat fluxes deepening and decoupling the PBL. As latent heat fluxes become larger than turbulence driven by radiative cloud-top cooling, the PBL deepens and turbulence no longer sustains mixing throughout the entire depth of the layer, causing the cloud and surface layers to decouple thermodynamically. Conditional instability builds at the surface, allowing shallow cumulus to form while the reduced moisture fluxes leads to stratocumulus breakup and dissipation. Precipitation can also lead to PBL decoupling. Evaporation of drizzle drops from stratocumulus cools and moistens the subcloud layer relative to the cloud layer, resulting in a stabilization of the boundary layer (Savic-Jovcic and Stevens, 2008). The stabilization can result in a weakened turbulent mixing and promote a transition to a more cumulus-dominated cloud field.

Extratropical cyclones, which dominate synoptic variability in the midlatitudes, exhibit much more transient and spatially heterogeneous cloud structures than those found in the subtropics, especially in winter (Wang et al., 1999; Wood et al., 2015; Mechem et al., 2018). The different regions of a cyclone are characterized by different meteorological conditions, and the PBL structure is strongly linked to the thermal advection patterns and associated large-scale vertical motion (Sinclair et al., 2010). Over the warm sector, the warm conveyor belt (WCB) transports warm, moist air ahead of the cold front poleward (Harrold, 1973) and results in negative surface heat fluxes, where energy is transported from the atmosphere to the ocean (Boutle et al., 2010; Sinclair et al., 2010). The warm air advection and convergence of the WCB are associated with ascent, high clouds, and precipitation in the warm sector (Houze et al., 1976; Boutle et al., 2010; Naud et al., 2015). The PBL is shallow ($\sim$500 m) and stable (not turbulent) (Sinclair et al., 2010), but shallow cumulus may still form (Field and Wood, 2007; Naud et al., 2015). The cold conveyor belt (CCB) transports cooler air ahead of the warm front westward (Carlson, 1980). Along the CCB and cold front, deeper clouds and precipitation form, contributing to the well known cyclone comma shape (Boutle et al., 2010; Field and Wood, 2007).

Behind the cold front, cold and dry advection results in positive sensible and latent heat fluxes, promoting a PBL that is well-mixed, unstable (turbulent), and deeper ($\sim$2.5 km) (Boutle et al., 2010; Sinclair et al., 2010). This region has the largest area fraction of PBL clouds, including both stratocumulus and cumulus (Field and Wood, 2007; Mechem et al., 2010; Wood,

2012; Rémillard and Tselioudis, 2015). Naud et al. (2018a) examined seven years of observations, yielding ∼1,800 cold fronts in the Eastern North Atlantic (ENA). They found that the post-cold-frontal region had stronger winds and subsidence compared to regions of northerly or northwesterly flow and subsidence that are far away and not associated with the cold front. Similarly, the post-cold front PBL is also drier, more unstable, more turbulent, and with larger surface fluxes, resulting in greater cloud cover and thicker clouds compared to similar flow and stability conditions not associated with a front. Kazemirad and Miller (2020) performed Lagrangian trajectory calculations from the stratocumulus to the broken cumulus regime in the cold sector and found that air moving southward experienced increased latent heat fluxes, resulting in more decoupled boundary layers and broken clouds, providing evidence that some cloudiness transitions in midlatitude cyclones are consistent with the mechanisms found in the subtropics (Bretherton and Wyant, 1997). Tornow et al. (2023) found that earlier onset of rain formation triggered earlier transitions from overcast to broken clouds during cold air outbreaks (CAO) in the cold sectors of midlatitude cyclones, similar to the precipitation-driven transitions in the subtropics (e.g., Savic-Jovcic and Stevens, 2008). In some situations, slantwise dry-air intrusions in the cold sector have been shown to additionally influence cloud and boundary layer properties through increases in surface fluxes and associated turbulence, which can deepen the PBL and increase the cloud cover on average (Ilotoviz et al., 2021; Tornow et al., 2023).

The coarse temporal and spatial resolution of the observational network is not able to resolve the detailed PBL structures in a synoptic cyclone (Ghate et al., 2011). While satellite imagery is able to characterize the spatial structure of the cloud field, the vertical structure of the PBL, inversion, and lower troposphere is not well sampled except near upper-air sounding locations. Therefore, we use regional models, which have much finer spatial and temporal resolution compared to observations, to study PBL clouds in midlatitude cyclones and CAO. However, despite their importance to local weather, marine and aircraft operations, and the global energy budget, both global and regional models struggle to accurately represent the boundary layer clouds and precipitation associated with midlatitude cyclones (Bodas-Salcedo et al., 2012; Field et al., 2014, 2017; Naud et al., 2020). This is in part due to only partially resolved shallow convection processes. In km-scale models, convection is assumed to be represented to some degree (convection "permitting" or "allowing" grid sizes), but important motions at the scale of convection, as well as smaller boundary-layer eddies, are not resolved, resulting in errors (Bryan et al., 2003). For example, Abel et al. (2017) simulated cold air outbreak (CAO) clouds and found that the Met Office Unified Model accurately represents the shallow cumulus regime but does not produce the stratocumulus regime. Broadly speaking, these models tend to underestimate the boundary layer cloud liquid water content, cloud fraction, and albedo (Abel et al., 2017; Bodas-Salcedo et al., 2012; Field et al., 2017; Nelson et al., 2016; Wyant et al., 2015).

However, model resolution is often not the only issue in play. Field et al. (2017) simulated a CAO and showed that as the mesoscale model resolution increased, the clouds became even more broken and the stratocumulus regime disappeared altogether, suggesting fundamental structural errors in the model representation of physics that were not ameliorated by better resolution. More sophisticated treatments of model physics have been yielding improvements to PBL forecasts. Zheng et al. (2024) was able to qualitatively simulate the overcast to broken cumulus transition using 3 km grid spacing in a global cloud

resolving model, including the increases in surface fluxes and deepening of the PBL which leads to stratocumulus breakup
and deeper shallow convection. However, the liquid water was still underestimated, and the mesoscale organization was not
well-represented compared to satellite. In summary, models continue to struggle representing boundary-layer cloud-regime
transitions, and cloudiness transitions across synoptic systems remain poorly understood, especially in the context of transition
hypotheses developed for subtropical cloud systems.

Understanding the factors driving the evolution of cloud and boundary layer properties in midlatitude cyclones is crucial for
improving numerical model forecasts. Here we apply the Naval Research Laboratory Coupled Ocean/Atmosphere Mesoscale
Prediction System (COAMPS, Hodur, 1997) regional model to a case study of a wintertime midlatitude cyclone over the East-
ern North Atlantic. We apply our understanding of boundary-layer clouds, largely relevant to barotropic atmospheres in the
subtropics, and expand it to boundary-layer clouds accompanying the midlatitude baroclinic synoptic system. We have two
linked objectives: 1. to evaluate to what extent COAMPS represents the synoptic controls on cloud regime and PBL evolution
within a midlatitude cyclone, including an analysis on the model's sensitivity to the microphysical parameterization; and 2. to
examine the transitions of synoptic, cloud, and boundary layer properties across different regions of the cyclone in both obser-
vations and COAMPS, in a way that minimizes errors associated with cyclone strength and phase. We discuss the observational
datasets used and modeling setup in Sect. 2. Section 3 discusses the synoptic setup of our case study. We present our model
evaluation results in Sect. 4. In Sect. 5, we analyze model output and observations of boundary layer and cloud properties
throughout the cyclone. Finally, in Sect. 6, we discuss the results and conclusions.

## 2  Data and Methods

### 2.1  Observations

We use the wealth of continuous observations taken at the Atmospheric Radiation Measurement (ARM) Eastern North Atlantic
(ENA) permanent atmospheric observatory deployed on Graciosa Island, Azores. The Azores straddle the transition between
subtropics and midlatitudes, with large-scale influences from both the Bermuda High and Icelandic Low. The winter season
exhibits the greatest synoptic variability, indicated by larger values of 500 hPa geopotential height standard deviations, and
cloud patterns are much more transient (Wood et al., 2015). A study that used self-organizing maps to characterize synoptic
states over ENA found that in winter, midlatitude cyclone centers tend to remain north of the island, but the associated cold
fronts drape across the Azores (see Fig. 8 in Mechem et al., 2018). Lying in the vicinity of the active cold-season storm track
makes the Azores region of the ENA an ideal environment to study transient cloud behavior associated with midlatitude cy-
clones. This study will focus on a midlatitude synoptic system impacting the Azores during 22-29 January 2018 during one of
two intensive observation periods (IOPs) that were part of the Aerosol and Cloud Experiments in the Eastern North Atlantic
(ACE-ENA) field campaign, taking place in summer (June-July) 2017 and winter (January-February) 2018 (Wang et al., 2022).
We note that January 2018 was in a positive phase of the North Atlantic Oscillation, so both the Icelandic Low and Bermuda
High were stronger than normal, which tends to shift the storm track northward (Hurrell, 1995). However, the cyclone in our

case study follows the average wintertime North Atlantic storm tracks as in Neu et al. (2013) and Wang et al. (2023).

We use the Active Remote Sensing of Cloud Layers (ARSCL) product that combines data from the 35-GHz Ka-Band ARM
Zenith Radar (KAZR), ceilometer, and lidar retrievals (Clothiaux et al., 2000; Johnson et al., 2014). The KAZR measures
the reflectivity and Doppler velocity of clouds and precipitation passing over the site and has a range resolution of 30 m and
temporal resolution of 4 s. We also employ the Synergistic Passive and Active Retrieval of Cloud (SPARCL, Cadeddu et al.,
2017, 2020, 2023), which combines KAZR and ceilometer retrievals using the methods in O'Connor et al. (2005). Soundings
were launched 4 times per day (every 6 h) during the IOP to quantify the thermodynamic profile of the atmosphere (Keeler
et al., 2022). The surface meteorological (MET) observations include the 3 m wind, 1 m pressure, and 1.25 m temperature at
1 min intervals (Kyrouac et al., 2021), and the 1290 MHz radar wind profiler (RWP) measures backscattered radiation, which
is used to determine the vertical profile of horizontal winds over 15 min averaging periods (Muradyan and Ermold, 2021).
Finally, the laser disdrometer measures hydrometeor size and fall velocity during precipitation (Zhu et al., 2014).

The SPARCL algorithm exploits the different scattering properties of small and large droplets measured from the 90 GHz
channel of the microwave radiometer in order to distinguish between cloud and drizzle water paths. The drizzle water paths
in the PBL clouds in this study are small ($<20 \mathrm{~g~m^{-2}}$), but only the cloud water paths are used regardless. Uncertainties in
the cloud LWP retrievals are on the order of $15 \mathrm{~g~m^{-2}}$ (Cadeddu et al., 2023). The subcloud evaporation is estimated as the
difference between the cloud base rain rate from SPARCL and the surface rain rate measured by the laser disdrometer, each
averaged over 15 min windows to allow the precipitation time to fall to the surface and to yield a robust statistical sample.
Cloud boundaries are identified from ARSCL, and cloud cover is estimated by the ground-based total sky imager (TSI, Flynn
and Morris, 2000; Long et al., 2001). Vertical profiles of liquid water potential temperature and total water mixing ratio are
calculated using the radiosonde temperature and moisture, LWP, cloud base height, and cloud top height to solve for the slope
of the liquid water mixing ratio profile (the adiabatic liquid water content lapse rate, $\Gamma_l$, see Eq. (4) of Zuidema et al., 2005).
The liquid water is assumed to be zero at cloud base and increases linearly by $\Gamma_l$ to cloud top.

The depth of the planetary boundary layer (PBL) is estimated by the height of the maximum value of the second deriva-
tive of the liquid water potential temperature profile ($\partial^2 \theta_l / \partial z^2$), which corresponds to the height of the maximum increases of
the $\theta_l$ gradient. Although we have not seen this approach used before, it was demonstrated to yield more consistent estimates of
PBL depth between observations and the model than methods based on the temperature gradient itself or Richardson number
thresholds. We calculate decoupling indices using the two methods described in Jones et al. (2011) based on the differences
between the upper PBL and lower PBL liquid water potential temperature ($D_\theta$) and total water mixing ratio ($D_q$) to infer the
amount of mixing and turbulence in the boundary layer. A PBL that is not decoupled ($D_{\theta,q} \leq 0.5$ K, kg kg$^{-1}$) is well-mixed
with a neutral thermodynamic profile, and a decoupled PBL ($D_{\theta,q} > 0.5$ K, kg kg$^{-1}$) has cloud properties that are thermo-
dynamically distinct from the surface because turbulence is not strong enough to mix through the entire layer. When we use
the term coupled or decoupled, we are referring to these specific thresholds defined in Jones et al. (2011), acknowledging that

in reality coupling and decoupling are a matter of degree and not a binary state. Finally, we calculate the estimated inversion strength (EIS) from Wood and Bretherton (2006). EIS is defined as $EIS = LTS - \Gamma_m^{850}(Z_{700} - LCL)$, where $LTS$ is the lower tropospheric stability (Klein and Hartmann, 1993), which is defined as $\theta_{700} - \theta_{1000}$. We use the potential temperature at 1000 hPa ($\sim$250 m) instead of the surface to avoid the strongly stable surface layers that are present throughout most of the study period. $\Gamma_m^{850}$ is the moist adiabatic lapse rate at 850 hPa, calculated from the 850 hPa temperature and moisture (equation 5, Wood and Bretherton, 2006). $Z_{700}$ is the height of the 700 hPa surface. Finally, $LCL$ is the height of the lifting condensation level. EIS measures the strength of the thermodynamic jump across the inversion and has been shown to have a positive correlation to cloud cover, when averaged over appropriate timescales, in both the subtropics (Wood and Bretherton, 2006; de Szoeke et al., 2016) and cold sectors of midlatitude cyclones (Naud et al., 2016).

Satellite images and analysis are from the Meteosat-10 geostationary satellite with 4 km resolution. Cloud property retrievals employ the Visible Infrared Solar-Infrared Split Window Technique (VISST) algorithm (Minnis et al., 2011) in which LWP is derived from the effective droplet size and visible cloud optical depth, and cloud top height (CTH) is estimated using the emissivity, cloud effective temperature, and a temperature profile obtained from global model output. The cloud coverage is the percentage of cloudy pixels within a $0.5 \times 0.5°$ box. These retrievals use visible imagery and therefore, the analysis is only performed using daytime data when the solar zenith angle is less than 65°.

The European Center for Medium-Range Weather Forecasts (ECMWF) Reanalysis v5 (ERA5, Hersbach et al., 2020) hourly atmospheric reanalysis dataset is also used to characterize the spatial patterns of clouds, horizontal wind, temperature, large-scale vertical motion, and surface fluxes surrounding the cyclone and as a check on the model simulation. ERA5 has a horizontal resolution of $0.25°$ latitude and longitude. Because the surface flux measurements at the Azores are taken over the island itself, we use the ERA5 surface fluxes over the ocean ($0.5°$ latitude north of the island) to provide representative values to evaluate against the COAMPS-calculated fluxes.

## 2.2 Configuration of COAMPS simulations

Simulations of the study period (22-29 January 2018) are performed using NRL COAMPS (Hodur, 1997). COAMPS is a mesoscale model based on non-hydrostatic, compressible dynamics. We use a doubly nested domain with two-way interaction, which allows us to perform moderately high-resolution simulations at a large spatial extent while still being computationally and operationally feasible. The coarse grid has a horizontal grid spacing of 9 km, $384 \times 384$ points, and a timestep of 10 s; the fine mesh has a horizontal grid spacing of 3 km and $397 \times 397$ points (Fig. 1a). The vertical grid uses 45 levels with varying grid spacing. In the boundary layer, the vertical grid spacing ranges from 20 to 80 m; the inversion layer spacing ranges from 80 to 150 m; and the spacing above the inversion ranges from 150 to 1000 m at 10 km (Fig. 1b).

COAMPS initial and boundary conditions are derived from the 6-hourly Global Forecasting System (GFS) 0-h analysis fields, which have an effective horizontal grid spacing of 27 km. GFS analysis and buoy observations are assimilated into COAMPS

using the 3D-variational data assimilation algorithm NRL Atmospheric Variational Data Assimilation System (NAVDAS). The weeklong simulation begins on 22 January 2018 at 00:00 UTC. The first 24 h of the simulation constitutes model spin-up, where observational data are assimilated every 12 h for the first 24 h (two total data update cycles). The model is then run without data assimilation for the next six days (144 h, until 29 January 2018 00:00 UTC), with forcing from the GFS only acting via the lateral boundaries on the COAMPS coarse mesh.

COAMPS uses the Kain and Fritsch (1990) parameterization for deep convective processes on the coarse mesh. The convective parameterization is not active on the fine mesh, because a 3 km grid lies in the realm of convective-permitting grid sizes (Prein et al., 2015; Weisman et al., 1997). The radiation parameterization is that of Fu and Liou (1992), and surface fluxes are parameterized as described in Louis (1979) and Wang et al. (2002). The boundary layer and turbulence processes are based on the level-2.5, 1.5-order Mellor and Yamada (1982) parameterization, and shallow convection is formulated based on Tiedtke et al. (1988). The shallow convection parameterization is active in both meshes.

The COAMPS operational microphysics parameterization uses the single-moment Rutledge and Hobbs (1983) formulation, with the warm-rain component based on Kessler (1969). This study is predominantly concerned with warm-rain (liquid only) processes. However, COAMPS produces mixed phase clouds with frozen particles making up about 10-50% of the total in-cloud mass in some of the colder convective clouds. Over and south of the Azores region, COAMPS produces very little ice. In situ observations of ice content in wintertime boundary layer clouds at the Eastern North Atlantic site were not a priority during the ACE–ENA field campaign, so it is uncertain whether or not these clouds contain ice. Though we do not expect this amount of ice to have a significant impact on the forecasts, an overestimation of ice content by the model could result in an underestimation of liquid water and less warm rain precipitation.

The Kessler (1969) scheme is simple with several arbitrary parameters (Morrison et al., 2020), whereas the Khairoutdinov and Kogan (2000) (KK2000 hereafter) has parameters derived from nonlinear regression of bin-microphysics spectra taken from large-eddy simulations of shallow clouds. Because of its greater physical representativeness, we supplement the Kessler (1969) operational warm-rain microphysics in COAMPS with elements from the KK2000 drizzle parameterization. The Kessler (1969) autoconversion, accretion, fall speed, and evaporation expressions are replaced with those from KK2000 to improve the representation of boundary-layer cloud precipitation processes without adding the complexity of a full double-moment scheme. The KK2000 autoconversion expression follows the functional form:

$$\left(\frac{\partial q_r}{\partial t}\right)_{auto} = c q_c^{\alpha} N_c^{\beta} \tag{1}$$

where $q_r$ is the rain water mixing ratio [kg kg$^{-1}$], $q_c$ is the cloud water mixing ratio [kg kg$^{-1}$], $N_c$ is the cloud drop concentration [cm$^{-3}$], and $c$, $\alpha$, and $\beta$ are constants, yielding an autoconversion rate with units [kg kg$^{-1}$]. KK2000 performed a regression analysis on simulated large eddy simulation stratocumulus drop spectra and found the constants to be: $c = 1350$,

$\alpha = 2.47$, and $\beta = -1.79$. The KK2000 accretion is:

$$\left(\frac{\partial q_r}{\partial t}\right)_{accr} = 67(q_c q_r)^{1.15} \tag{2}$$

The fall speed is:

$$V_{q_r} = 0.12 r_{vr} - 0.2 \tag{3}$$

where $r_{vr}$ is the mean volume radius of the drizzle drops [μm] and is estimated as:

$$r_{vr} = \left(\frac{4\pi\rho_w}{3\rho_a}\right)^{-1/3} q_r^{1/3} N_r^{-1/3} \tag{4}$$

where $\rho_w$ is the density of water and $\rho_a$ is the air density. This yields terminal velocities with units $[\text{m s}^{-1}]$. Finally, the evaporation is:

$$\left(\frac{\partial q_r}{\partial t}\right)_{evap} = 3C_{evap}G(T,p)\left(\frac{4\pi\rho_w}{3\rho_a}\right)^{2/3} q_r^{1/3} N_r^{2/3} S \tag{5}$$

where $C_{evap} = 0.55$ and $G(T,p)$ is from the drop radius growth equation as found in Yau and Rogers (1996) or elsewhere. We refer to our implementation of this microphysics parameterization as "KK Lite" because we do not run a full double moment scheme. Rather, we assume a constant cloud ($N_c$) and drizzle ($N_r$) drop concentration for the entire simulation. Our control simulation employs the KK Lite parameterization and assumes an $N_c$ of 100 $\text{cm}^{-3}$. This value is broadly representative of maritime conditions, although measurements of cloud condensation nuclei (CCN) at Graciosa Island over the simulation period range from 40 to over 400 $\text{cm}^{-3}$ (0.2% supersaturation). We assume an $N_r$ of 0.01 $\text{cm}^{-3}$, a value based on bin-microphysics large-eddy simulation (LES) findings in marine low clouds (specifically, the System for Atmospheric Modeling – Explicit Microphysics (SAMEX) results from a number of studies across different PBL cloud regimes (vanZanten et al., 2011; Mechem et al., 2012; Rémillard et al., 2017)). Although most single-moment microphysical parameterizations functionally relate $N_r$ to the mass field ($q_r$), we show in Sect. 4 that KK Lite displays only modest sensitivity to the constant values of $N_r$ we choose. Equations (3) and (4), along with $q_r$, are used to calculate the instantaneous, grid-scale cloud base and surface rain rates in COAMPS. The subcloud evaporation is calculated as the difference between the cloud base and surface rain rates.

The internally computed COAMPS PBL height is determined by a Richardson number threshold of 0.5. The Richardson number approach is robust but tends to underestimate the inversion height by at least 200 m (Wang et al., 2011; Wyant et al., 2010) in cases of a pronounced inversion. Gradient methods have been shown to be more accurate than Richardson number approaches in both soundings and models (e.g., Liu and Liang, 2010; Eleuterio et al., 2004). As described in Sect. 2.1, instead of using the Richardson number PBL heights, we estimate PBL depth for both COAMPS and the soundings by the height of the maximum in the second derivative of the liquid water potential temperature. As in the observations, we calculate additional parameters using the thermodynamic profiles, including the inversion strength (EIS) and decoupling indices. Calculations of cloud fraction in COAMPS include a subgrid-scale contribution that is parameterized as a function of relative humidity (Slingo,

1987). At grid spacings of $\sim$3 km, this formulation is is predominantly all-or-nothing (i.e., grid-point cloud fractions of either

0 or 1), but it does allow for non-zero cloud fractions at relative humidity values above 70%, which is of increasing importance

for larger grid sizes. For calculations of cloud cover, we use the maximum cloud fraction between 300 and 2000 m at each grid

point (maximum overlap assumption).

## 2.3   Sensitivity experiments

In addition to the baseline COAMPS simulation described above using the KK Lite parameterization, we perform a series of

sensitivity experiments. These serve to compare our baseline to the operational Kessler (1969) warm-rain microphysics and the

Thompson et al. (2008) parameterization, which is a currently available option in COAMPS. We also evaluate the sensitivity

to results to variations in $N_c$ and $N_r$, specifically with experiments specifying $N_c$ and $N_r$ values of 40 cm$^{-3}$ and 0.1 cm$^{-3}$,

respectively. We also conduct a simulation of KK Lite with both $N_c$ and $N_r$ changed to evaluate the mutual interactions among

the two parameters (Stein and Alpert, 1993). The final sensitivity simulation tests the sensitivity to the vertical grid resolution

by using an 87-point grid with higher resolution (25 m) throughout the entire boundary layer (0-2 km), though this has no

significant impact on the forecasts and results are not shown. A detailed list of the sensitivity experiments is in Table 1.

In addition to the simulations listed in Table 1, we perform another series of COAMPS simulations using the control KK

Lite configuration to quantify statistical forecast error. Those simulations all include a 24-h spin-up period, followed by a run

of varying lengths. The subsequent simulations begin 12 h after the previous and all finish 29 January 00:00 UTC. This yields

12 total simulations, with twelve 12-h forecasts, eleven 24-h forecasts, ten 36-h forecasts, and so on.

## 2.4   Cyclone center and frontal identification

The most novel aspect of this study is the analysis framework of tracking the cyclone center and associated cold front. The

center and fronts are identified in COAMPS using the Bauer et al. (2016) and Naud et al. (2016) algorithms, respectively. The

Bauer et al. (2016) algorithm identifies and tracks cyclones using local minima in SLP, which are then filtered by a threshold

based on topography, latitude, and season to ensure that the cyclone identified is meaningful. Once cyclone centers are iden-

tified, associated fronts are located based on both the 1 km horizontal potential temperature gradient (Hewson, 1998) and 6

h change of the 850 mb wind direction and magnitude (Simmonds et al., 2012). Naud et al. (2016) applies several filters to

the frontal locations found by the Hewson (1998) and Simmonds et al. (2012) methods to find the best estimate of warm and

cold front locations. The cyclone centers and associated fronts are then tracked over time at 6-h increments (Bauer et al., 2016).

A cold-front-centered compositing approach (e.g., Field and Wood, 2007) is used to transform the model output into a cold-

front-centered coordinate system. This technique highlights the differences between pre-frontal, frontal, and post-cold front

environments and allows for straightforward comparisons among the three regions. This system-relative analysis framework

also serves to minimize any phase error the model may have associated with a too slow or too rapid movement of the system.

We analyze transects across the cold front to explore the joint variability of cloud and boundary-layer properties (LWP, PBL depth) and meteorological conditions (stability, vertical motion, surface fluxes).

## 3 Description of case study

Figure 2 shows ERA5 low cloud cover (between approximately 800–1000 hPa) and sea level pressure (SLP) over the ENA every 24 h at 0000 UTC throughout the simulation period. During the first 48 h of the study period, a low-pressure center is located about 17° north of Graciosa Island. By 0000 on the 24th, the low becomes zonally elongated, with the associated cold front extending south and southwest from the eastern part of the low center (Fig. 2b). The frontal locations in Fig. 2a-c are based on the automated frontal identification algorithm, described in Sec. 2.4 and applied to ERA5 data. The cyclone appears to be occluded throughout the period, with the cold fronts originating from the secondary low formed from the frontal instability of the triple point (Schemm and Sprenger, 2015). The Bermuda High center is situated just west of the Azores (Fig. 2b). Sky conditions are nearly overcast along the cold front and at the center of the high, while broken boundary layer clouds are present behind the cold front (Fig. 2c). By 26 January, the center of the high is located over the Azores (Fig. 2d) and continues to move eastward over the final three days of the case study period (Fig. 2e,f).

Observations on Graciosa Island during the first 24 h of the study period are characterized by southwesterly winds, broken clouds and a relative minimum in 500 hPa geopotential height and surface pressure (Fig. 3). A cold front passes over ENA around 00:00 UTC on 24 January, consistent with the ERA5 front identification (Fig. 2b). After the cold front passage, surface pressure increases, the temperature continues to decrease, and winds change from southwesterly to northwesterly (Fig. 3b,c,d). The KAZR observed precipitating convection along the cold front, and broken, drizzling cumulus with tops at 1.75 km after the cold front passage (Fig. 3e,f). Around 25 January 18:00 UTC, the surface temperature begins to increase, and the winds change to have a southerly component with the eastward propagation of the high-pressure system (Fig. 3c,d). The clouds transition from broken cumulus to an unbroken sheet of periodically drizzling stratocumulus with a cloud-top height as great as 1.8 km (Fig. 3e). The unbroken stratocumulus persists for nearly 48 h, within which the cloud top gradually decreases to 1.3 km. On 28 January, the stratocumulus begins to transition to broken cumulus (the low cloud bases in Fig. 3e).

When the winds have a southerly wind component (90°–310°; between the green bounds in Fig. 3d), the ENA thermodynamic, cloud, and precipitation quantities may include influences from the island instead of being representative of pure marine conditions (Ghate et al., 2021). Furthermore, the conditions during this period are dominated by the subtropical Bermuda High as opposed to the synoptic cyclone. For these reasons, our evaluation of COAMPS will emphasize the earlier period with northerly wind conditions (24 January 00:00 UTC–25 January 18:00 UTC). Follow-up efforts will include exploration of the transition to the southwesterly flow regime.

## 4 Evaluation of COAMPS simulations

We evaluate COAMPS against observations at ENA in a variety of ways. Comparing simulation output, which varies in time and space, to continuous observations at a single location is difficult. Discrepancies in cloud structures, and cyclone strength and position between the temporal and spatial analysis windows (phase error) may occur. We have chosen to define the COAMPS analysis region as the 144 km × 144 km (48 × 48 points) box centered on Graciosa Island (small green box in Fig. 1) over which we calculate statistics. The wind speed during the case period is at maximum $10 \mathrm{~m~s^{-1}}$, which corresponds to 36 km of advection in 1 h. Therefore, our COAMPS analysis box roughly corresponds to a 4-h observational window.

We quantify COAMPS forecast uncertainty by comparing 500-hPa geopotential height forecasts at various lead times against ERA5 reanalysis (Fig. 4). The forecast error for each simulation described in Sect. 2.3 is the mean absolute error (MAE) between ERA5 and all COAMPS coarse mesh points regridded to the ERA5 grid. The MAE of the forecast errors at each forecast time is evaluated. As expected, the forecast errors increase as the forecast time increases. However, COAMPS appears to be representing the large-scale meteorology well, with SLP errors of less than ∼1 hPa and 500 hPa height errors of less than 25 m throughout the entire domain and out to 72 h (Fig. 4). Errors increase at a small rate (or even decrease for the 72-h simulation) potentially due to the dominance of the slowly evolving high pressure system in the later parts of the simulation period. The small errors in meteorology suggest that the model biases discussed in the following paragraphs cannot be attributed to errors associated with the cyclone strength or phase.

We next determine how well COAMPS represents the liquid water potential temperature and water vapor mixing ratio profiles compared to the soundings as well as cloud boundary evolution compared to ARSCL cloud boundary retrievals (Fig. 5). For nearly all of the northerly wind period following the passage of the cold front (24 January 11:30 UTC–25 January 11:30 UTC), COAMPS underestimates the temperature profile throughout the lowest 3 km of the atmosphere. Over the same period, COAMPS slightly overestimates the boundary-layer moisture but underestimates the free-tropospheric moisture. COAMPS also underestimates the inversion height and the temperature gradient across the inversion, potentially due to the coarse vertical resolution of the model at this height. The slightly cooler and moister boundary layer results in a cloud base that is too low and a cloud that is too thick, all consistent with an underestimate of entrainment rate. During the southerly wind conditions (after 25 January 17:30 UTC), the boundary layer temperature and moisture are closer to the observed profiles (with the exception of the moisture at 17:30 UTC). For the last two sounding periods, COAMPS overestimates the base of the inversion and again fails to produce a sharp inversion. This results in higher cloud tops and bases than observed. Here, the underestimate of moisture, overestimate of inversion heights, and weak inversion strengths are consistent with an overestimate of entrainment.

Although Rémillard et al. (2012) showed that ENA is in a near constant state of decoupling, the radiosondes only indicate a decoupled PBL ($D_{\theta,q} > 0.5$ K, g kg$^{-1}$) for the first 24-30 h after the cold front passage. While the decoupling index values for COAMPS and the soundings both indicate decoupling over most of this period, the vertical structure of some of the

COAMPS temperature and moisture profiles differs substantially from the observations (Fig. 5). The reason for the larger COAMPS indices is because the coarse model resolution causes the top of the PBL to be above the inversion and therefore the upper PBL layer is warmer and drier than the surface layer. The discrepancy between the ARSCL and COAMPS cloud base heights is also consistent with the discrepancy in the degree of decoupling. In coupled boundary layers, the lifting condensation level (LCL) and cloud base are at nearly the same height, but in decoupled boundary layers, they may diverge by several hundred meters due to drier air in the cloud layer (Jones et al., 2011). In the soundings, entrainment appears to warm and dry the cloud layer compared to the surface, resulting in a decoupled cloud layer that is ∼300 m above the LCL (not shown). In COAMPS, because of the poorly represented inversion, the air entrained into the boundary layer is much cooler relative to the soundings and therefore does not decouple the cloud and subcloud layers. Rather, the boundary layer remains closer to well-mixed conditions, resulting in a cloud base that is near the LCL. After 30 h, the height of the inversion steadily decreases, and both COAMPS and the soundings maintain a well-mixed profile for the remainder of the northerly wind period, suggesting that the PBL thermodynamic structure in cyclones is highly transient compared to ENA conditions during more quiescent, periods (Rémillard et al., 2012).

Low-level winds can also play a role in PBL evolution, so we compare COAMPS, sounding, and wind profiler (RWP) wind speeds throughout the period (Fig. 6). COAMPS represents the low-level wind speeds credibly compared to observations, but slightly overestimates the near-surface (<500 m) wind speed and underestimates the upper-level wind speed. Discrepancies in the representation of the PBL structure between COAMPS and the soundings (Fig. 5) do not appear to be attributable to the PBL winds. Boundary-layer winds are relatively constant with height and weaken throughout the period. The gradual decrease in wind speeds and boundary-layer turbulence corresponds to a gradual decrease in boundary-layer depth further away from the cold front, as expected.

Figure 7 compares bulk sounding parameters between COAMPS and seven observed soundings during the northerly wind period. On average, all of the COAMPS microphysics sensitivity experiments underestimate the inversion strength (EIS) but predict it within ∼1 K (Fig. 7a). The Kessler and KK Lite simulations underestimate the PBL depth by 138 m (8%) and 104 m (6%), respectively (Fig. 7b), and aside from one outlier, the depths are predicted within 100 m. The Thompson simulation overestimates the depth, on average, by 163 m (10%) (Fig. 7b). The underestimate of PBL depth reflects well-known, long-standing bias in regional (and larger-scale) models underestimating the depth of the marine PBL inversion (Wyant et al., 2010, 2015) that is predominantly tied to how poorly the model vertical grid represents the inversion. In our case, this error seems reasonable given the vertical grid spacing of ∼100 m at this height, and suggests our method for calculating PBL depth is a substantial improvement on the Richardson number method. Although we have tested a higher resolution vertical grid, grid spacings of 25 m across the inversion are still too coarse to result in significant improvement of the PBL depth. All COAMPS simulations, on average, overestimate the decoupling index and struggle to represent the degree of decoupling, with errors ranging from 25-100% (Fig. 7c). As described above, this is likely because the model inversion strength is too weak. However, Kazemirad and Miller (2020) used a threshold of $D_q < 1.6$ g kg$^{-1}$ to indicate not-decoupled profiles, making all COAMPS and observed

profiles not decoupled according to their metric. We characterize the differences in MAE values across the simulations (0.09 K EIS; 60 m for PBL depth; and 0.18 for the decoupling index) as only modest sensitivity to the microphysical parameterization. More precisely, the simulations exhibit a degree of bias relative to the observations, but that bias is only minimally attributable to differences among the microphysical parameterizations. Below we provide an explanation for the differences among the microphysical sensitivity simulations.

Probability distribution functions (PDFs) from COAMPS and the observations are compared over the northerly wind period in Fig. 8. The ENA observations have high temporal resolution but at a single point, so we attempt to find the temporal window over which the observations correspond to a finite spatial domain size. COAMPS PDFs are created from forecast hour period 32 h–64 h and over a 144 km $\times$ 144 km box centered on Graciosa Island. The observed PDFs are created from a temporally equivalent sampling window, accounting for advection (assuming a 10 m s$^{-1}$ wind speed), between 24 January 2018 00:00 UTC to 25 January 2018 18:00 UTC (COAMPS forecast hour 30-66). The observational sampling window is 4 h ($\pm$ 2 h) longer than COAMPS to account for advection (assuming a 10 m s$^{-1}$ wind speed), through the COAMPS analysis box. All variables are conditioned on nonzero elements. In addition, Table 2 compares averages of the variables over this 144 $\times$ 144 km$^2$ ENA region during the northerly wind period of each microphysics parameterization sensitivity simulation.

Relative to the observations, we find that all three COAMPS simulations produce fewer clouds at small LWP values (50-100 g m$^{-2}$) and more, large-LWP clouds (>200 g m$^{-2}$) (Fig. 8a). With the exception of the Kessler simulation (55.3 vs 54.2 g m$^{-2}$ in Table 2), all of the others considerably overestimate the mean LWP values at the ENA site. In addition, all simulations overestimate the cloud thickness, producing clouds 50-200 m too thick (Table 2). COAMPS also underestimates clouds at low cloud fractions (<0.25) and overestimates cloud fractions of 0.75-1.0 (Fig. 8e), potentially missing some of the broken shallow convective clouds due to the coarse horizontal resolution and instead producing grid-scale convection. Observations of precipitation rate at cloud base and the surface show that nearly all of the precipitation evaporates before reaching the surface (Fig. 8b,d, and Table 2). The COAMPS simulations, on the other hand, considerably underestimate the cloud base rain rate and largely overestimate the surface rain rate, resulting in a large underestimation of subcloud evaporation. The KK Lite control simulation produces the largest cloud base rain rate relative to the Kessler and Thompson parameterizations, owing to the suitability of its autoconversion, accretion, and fall speed relations for marine low clouds. The PBL depth is predicted to within 100 m for all simulations (Table 2).

Means over the entire fine mesh domain are compared in the italicized numbers in Table 2 to describe the sensitivity between the different simulations. Many but not all of the differences between simulations are consistent with the variations in microphysical parameterizations and assumed parameters. The fraction of the cloud-base precipitation that evaporates depends on the subcloud humidity and the shape of the drop-size distribution, with a larger number of smaller drops having greater total surface area and are therefore easily evaporated. For this reason, the KK Lite simulations with $N_r = 0.1$ cm$^{-3}$ exhibit a larger fraction of precipitation that is evaporated relative to the other KK Lite simulations with $N_r = 0.01$ cm$^{-3}$. The smaller

magnitudes of cloud-base precipitation for the $N_r = 0.1$ cm$^{-3}$ simulations are consistent with the smaller raindrops associated with a larger value of $N_r$ having slower sedimentation velocity. Higher evaporation rates result in increased boundary-layer

stability (Nicholls, 1984), reduced turbulence and entrainment, and a decreased PBL height (Stevens et al., 1998). The Thompson simulation produces the smallest evaporation rates and subsequently has the highest PBL height. However, both KK Lite parameterizations with $N_r = 0.01$ cm$^{-3}$ have the largest subcloud evaporation rates but not necessarily the shallowest PBL heights, and the Kessler simulation has a similar evaporation rate as Thompson but PBL depths that are much shallower than Thompson and closer to the KK Lite simulations. These discrepancies indicate that other factors besides microphysics, such

as surface fluxes or large-scale vertical motion, are acting alongside microphysical processes to impact PBL depth. These results show that although COAMPS does not exactly reproduce the observations, it is largely internally consistent with the relationships found in observations.

## 5   Boundary-layer cloud properties composited on synoptic features

The objectively identified cold front from the COAMPS simulation at 06:00 UTC on January 24, 2018, just after the time of

the cold front passage over the Azores is shown in Fig. 9. The identified cold front lies in a reasonable position, judging from the location of the cold front on the 1 km potential temperature field. The frontal location is also very similar to the location identified using ERA5 data (similar to the Fig. 2b front from six hours prior). Figure 9 also shows the location and width of the stationary transect we use for the analysis in this section. The transect passes through Graciosa Island and is perpendicular to the cold front. A transect width of ∼111 km is created, and variables parallel to the cold front are averaged along the transect.

This approach of a rectangular analysis area incorporates more cloudy and precipitating grid points compared to a single line.

We separate grid points in the fine mesh and along the transect between the cold, frontal, and warm sectors. The frontal sector is distinguished as the 150 km ahead of (to the southeast) and behind (to the northwest) the cold front, identified to include the upward vertical motion associated with the front. This also corresponds to the distance of influence that the cold

front has on the boundary layer, assuming a PBL depth of 3 km (a generous value for the PBL depths in this study) and a frontal slope of 1:50. The cold sector is defined by all points to the northwest of the frontal region, and the warm sector includes all points ahead (to the southeast) of the frontal region. The synoptic and thermodynamic variables are composited onto a cold-front-centered framework for eight model forecast times 6 h apart, starting with forecast hour 12 and ending with 54. At hour 12 of the simulation, most of the transect is ahead of the cold front and, therefore, mainly samples the properties of the

warm sector. In this case, the warm sector region is far south of the warm front, so the transect captures the properties of the warm sector but not those associated with the warm front itself. As the cyclone moves over time, the transect captures more of the area further behind the cold front.

Figure 10 shows a map of COAMPS grid-scale cloud-base vertical velocities and LWP in the fine mesh. Clouds along and

ahead of the cold front are organized into a banded structure associated with weak upward motion (Fig. 10c,e). Satellite-

derived LWP also shows the banded structure of the clouds along and ahead of the cold front (Fig. 10a). COAMPS produces cellular clouds behind the cold front, which have high LWP and are associated with strong updrafts in the center with downdrafts on the cell edges, reminiscent of closed-cellular stratocumulus clouds (Fig. 10d,f). However, the satellite image suggests that the cold sector clouds are more reminiscent of open cellular convection, with clear centers and cloudy cell walls (Fig. 10b).

The LWP (and associated cloud thickness) as well as the horizontal size of individual cells in COAMPS are much larger than in the satellite, which was also found compared to the ground-based remote sensing observations in section 4 (Fig. 8 and Table 2). This results in a cloud cover that is much less than the satellite. The inversion strength parameter (EIS) has been shown to correlate well with cloud cover in both the subtropics, when averaged over appropriate timescales (Wood and Bretherton, 2006; de Szoeke et al., 2016), and in the cold sectors of synoptic systems (Naud et al., 2016). In this study, COAMPS un-

derestimates the inversion strengths (Figs. 5, 7), which may help explain the lack of COAMPS cloud cover. The inability of COAMPS produce sufficient drizzle and decouple may help explain the overestimation of LWP. Furthermore, the 3 km horizontal resolution may be too coarse to resolve the individual cells that are shown in the satellite image. However, we interpret the COAMPS cloud features as the model's attempt at mesoscale cellular organization despite the somewhat coarse grid spacing, as in Mechem and Kogan (2003). In the northwest corner of the domain, the cellular structure transitions to complete overcast

conditions, with clouds reminiscent of stratus or stratocumulus (Fig. 10d). These more horizontally homogeneous clouds are associated with near-zero or weak downward motion (Fig. 10f).

Figure 11 shows the median vertical cross sections, composited in 100 km wide bins, across the transect. The boundary layer in the warm sector is moist and conditionally unstable (i.e., not well-mixed), reminiscent of the trade wind cumulus

environment, and is characterized by periods of weak ascent and descent. The frontal region has sloped potential temperature and vapor mixing ratio profiles and a narrow band of weak ascent in the lowest 1 km that is sloped towards the cold sector with height. Behind the cold front, the boundary layer is ~10 K cooler, ~5 g kg$^{-1}$ drier, and well-mixed with inversion heights of ~1.5 km just behind the front and 1.2 km 1500 km behind the front. Subsidence dominates the boundary layer, inversion, and free tropospheric layers throughout the entire cold sector region, as expected from Naud et al. (2018a) and consistent with the

large-scale divergence found in the cold sector by Field and Wood (2007). The deep and turbulent boundary layer capped by a temperature inversion which is sustained by large-scale subsidence is comparable to the environment of the eastern portions of subtropical high-pressure systems where stratocumulus generally form.

COAMPS LWP values ahead of the cold front in the warm sector are near zero with only a few instances of broken clouds (Fig.

12a). The cloud cover over the warm sector can be as large as 40%, but this largely represents the subgrid-scale contribution, and the resolved cloud cover is only 0-10%. In the frontal region, clouds associated with the frontal convective band have LWPs that range from 50-150 g m$^{-2}$. The 750 km immediately behind the cold front have the largest LWPs with about 50% resolved cloud cover and 80% subgrid scale cloud cover, on average. Further behind the cold front, the LWP is lower, but cloud cover is nearly 100%. This provides evidence that COAMPS is trying to represent the observed cloudiness transition between

broken, stronger convective clouds close to the front and closed-cellular stratocumulus clouds further behind and closer to the

high (Fig. 3).

SSTs are 288 K in the northern part of the fine mesh and increase southward to 293 K in the southern portion of the fine mesh (not shown). This leads to sensible heat fluxes of <5 W m$^{-2}$ in the warm sector where warm air flows over relatively warm water, increasing into the cold sector to 50-150 W m$^{-2}$, where cold air flows over relatively warm water (Fig. 12b). The latent heat fluxes are over double the magnitude of the sensible heat fluxes and follow the same pattern, ranging from 10-90 W m$^{-2}$ in the warm sector and increasing to as much as 300 W m$^{-2}$ 750 km behind the cold front. Both the sensible and latent heat fluxes in this case are much stronger than in other studies of boundary-layer clouds over the subtropics and midlatitudes. LES intercomparisons for two DYCOMS-II cases yielded sensible and latent heat fluxes of 15 and 115 W m$^{-2}$ (RF01, Stevens et al., 2005), and 16 and 93 W m$^{-2}$ (RF02, Ackerman et al., 2009), respectively. An LES study of stratocumulus over the southeast Pacific during the VOCALS-Rex field campaign calculated sensible and latent heat fluxes of 4-8 and 55-70 W m$^{-2}$ (Mechem et al., 2012). Over the North Atlantic, sensible and latent heat fluxes during the ASTEX campaign ranged from -15 to +15 and 100-150 W m$^{-2}$ (Bretherton et al., 1995) in largely quiescent conditions dominated by the Bermuda High and trade winds (Albrecht et al., 1995). The larger surface fluxes in our study, particularly associated with the cooler air in the post-cold-frontal regions, suggests that cyclones substantially influence the surface fluxes, and the cold sector is much more closely related to the environments of cold air outbreaks (CAO) than the environments in which trade wind cumulus form. Li et al. (2022) find surface sensible and latent heat fluxes associated with CAO of 231 W m$^{-2}$ and 382 W m$^{-2}$, respectively, which is similar to those of this study. The peak in surface fluxes about 750 km behind the cold front is also associated with a maximum in LWP, in agreement with CAO studies which also find a transition from closed cell to open cellular clouds as the air-sea temperature difference and PBL turbulence increases (McCoy et al., 2017; Naud et al., 2018a).

The inversion strength and PBL depth generally follow the pattern of surface fluxes, with shallow PBL depth and weaker inversion stability ahead of the cold front and increasing PBL depth and inversion stability behind the cold front (Fig. 12c). The strong subsidence in the cold sector helps to maintain the strong inversions. However, despite this strong subsidence, the boundary layer remains deep, suggesting that the PBL depth is maintained by the strong surface fluxes (Sinclair et al., 2010) and associated shallow convection, a process that could be enhanced by dry intrusions (Raveh-Rubin, 2017). Precipitation is sporadic and most likely to occur within 750 km behind the front (Fig. 12d). The boundary layer is decoupled in every sector, with larger decoupling indices found in the warm and frontal sectors and gradually decreasing further behind the cold front. This gradual decrease in decoupling index towards a well-mixed PBL as the PBL depth decreases was also found in the soundings (Fig. 5), even though the COAMPS indices have a bias towards larger values. The largest values of decoupling index in the cold sector are within 500 km of the front, downwind from or where the majority of the precipitation occurs, as expected.

To further generalize the findings from the transects, we create box and whisker plots of the properties in each sector along the transects (Fig. 13). This figure highlights the drastic differences between the quiescent environmental conditions in the warm sector and the much more turbulent and strong-inversion conditions of the post-cold frontal environment. Compared to the

COAMPS warm sector, the cold sector has much deeper PBLs, a stronger inversion, and greater surface sensible and latent heat fluxes. The frontal sector is associated with the values of PBL depth, inversion stability, and surface fluxes in between those of the warm and cold sectors. The distribution of COAMPS properties from each sector are compared to ENA observations. COAMPS represents the properties of each sector relative to the other sectors correctly, with the largest median values of all four variables in the cold sector, smallest median values in the warm sector, and the frontal sector distribution somewhere in between. The PBL depth discrepancies between COAMPS and the observations are likely due to the gradient method not being a good measure of the PBL depth in the warm and frontal sectors where the PBL is stably stratified and strongly decoupled (e.g., Seibert, 2000).

Box and whisker plots of the cloud properties in each sector are also presented (Fig. 14). COAMPS produces the widest range and highest LWPs in the cold sector, with values sometimes exceeding $500 \text{ g m}^{-2}$, and the smallest LWPs in the warm sector. COAMPS cloud top heights in the cold and frontal sectors are very similar, ranging from about 1.5-2 km, and are much higher than the warm sector cloud-top heights, which are only around 500 m. The cold sector also has the largest cloud coverage, ranging from about 60-100%. Only half of the warm sector has a cloud coverage above zero. ENA observations show that COAMPS overestimates the LWP (Fig. 8), but throughout the entire cold sector (Fig. 14a), satellite LWP retrievals show that COAMPS reproduces the distribution of clouds well, with only slightly too many high-LWP clouds, compared to the observations. Satellite retrievals indicate that the COAMPS cloud cover is too large (Fig. 14c), with the observations suggesting a more broken cloud field. On the other hand, we previously showed that that resolved-scale cloud coverage is underrepresented (Fig. 10, 12a), suggesting that the subgrid-scale cloud fraction parameterization is too aggressive when adding cloud cover to subsaturated regions. COAMPS also underestimates the LWP, height, and coverage of warm sector clouds, which probably is largely tied to the struggle with representing broken cloud fields on a 3-km horizontal grid. Overall, these results are encouraging that COAMPS is better able to represent the boundary-layer and cloud properties in this system-relative context as compared to what the point-by-point comparisons with the observations would suggest.

To illustrate the relationship between cloud properties and variables representative of large-scale environmental conditions, we average $144 \times 144 \text{ km}^2$ boxes throughout the fine mesh at 6-h intervals between forecast hour 12 and 54, ending when the post cold front region begins to include areas of southerly winds. These regions are then separated into cold, frontal, and warm sectors. Figure 15 plots the mean liquid water content for each of these regions in a parameter space of inversion strength (EIS) and 2500 m vertical motion, taken in the free troposphere and which we take to represent the large-scale vertical motion. Warm-sector clouds are characterized by small LWP and are associated with weak inversion stability and vertical motion. We take these to be the model's attempt at representing cumulus cloud fields (While the EIS can be below zero in the warm sector, it doesn't mean that the profile is absolutely unstable. Rather, the negative EIS is likely because many profiles in the warm sector are conditionally unstable with little to no capping inversion and weak lower tropospheric stability (LTS, Klein and Hartmann, 1993), making the method inapplicable in these cases). Clouds in the frontal sector are associated with weak stability and form in areas of both ascent and descent associated with the frontal circulation. They have much higher liquid water contents than

clouds over the warm sector. Cold-sector clouds characterized by large values of LWP tend to be located just behind the frontal zone. Like the frontal clouds, these post-frontal clouds with large LWP can occur in any vertical motion condition and are associated with weak inversion stability, which allows the partially resolved cloudy updrafts to grow to sufficient depth to yield large values of LWP. Further behind the cold front, the EIS is larger, and the clouds are in an environment dominated by large-scale subsidence, reminiscent of a stratocumulus-topped boundary layer, and consistent with influences from dry intrusions. Although the boundary-layer clouds do cluster into certain environments (i.e., stronger clouds with larger LWP clustering in areas weak stability and clouds with low LWP forming in high stability and subsidence), there is no meaningful statistical relationship between them. These findings support the conclusions of Myers and Norris (2013) and de Szoeke et al. (2016) of no instantaneous relationships among boundary-layer cloudiness, stability, and vertical motion.

## 6  Conclusions

Most of our understanding of boundary-layer clouds comes from observational and idealized studies over the subtropical and trade regions, yet boundary-layer clouds are also prevalent in midlatitude baroclinic synoptic systems. In this study, we use the COAMPS regional forecast model to explore low cloud properties and behavior during a multi-day period in the warm sector, cold-frontal region, and post-frontal cold sectors of a midlatitude synoptic system over the Eastern North Atlantic (ENA). We have composited synoptic, cloud, and boundary-layer properties from COAMPS output and observations relative to the cold front identified by the Naud et al. (2016) automated frontal detection algorithm. We demonstrate that this provides a pathway to compare boundary-layer cloudiness forecasts against observations in the context of the different sectors of midlatitude cyclones and in a manner that minimizes errors associated with misplacement of the cyclone center and frontal structures (i.e., phase error). Although the COAMPS forecast does not exactly match point-to-point with the observations at ENA, COAMPS somewhat credibly represents the different cloud regimes found over ENA, including the cloudiness transitions and associated differences in thermodynamic properties across the front.

Our main findings are as follows:

– The thermodynamic and dynamic properties of the cold and warm sectors differ dramatically, with cold sectors exhibiting much stronger subsidence, inversion stability, and surface fluxes, and a deeper boundary layer topped by thicker clouds and greater cloud cover, relative to the warm sector;

– Most of the boundary-layer clouds in the COAMPS simulations are associated with grid-scale vertical motions that are larger than individual clouds but reminiscent of the mesoscale cloud structures (cloud ensembles) observed in the case;

– Clouds in the post-cold-frontal regime exhibit similar properties to clouds in cold air outbreaks;

– Model biases in cloud properties relative to observations are not explained by differences in microphysics parameterization.

This study has revealed the synoptic, boundary-layer, and cloud properties along a composite transect from far ($\sim$1500 km) behind the cold front, across the cold front, and into the warm sector. In the cold sector, a continuum of cloud and boundary-layer properties exist, as in the subtropics. The environment far behind the cold front is most reminiscent of the region of the eastern subtropical ocean basins and is characterized by strong subsidence ($\sim$3 cm s-1) and stability across the inversion (EIS of $\sim$11 K). Moderate values of surface heat fluxes (150 W m$^{-2}$ latent and 50 W m$^{-2}$ sensible) lead to moderate PBL depths ($\sim$1.3 km). The lower PBL depths in this region allow for the boundary layer to be more readily coupled (i.e., nearly well-mixed) than any of the other regions. This region, like the subtropics, has nearly homogeneous cloud cover. In our study with 3 km horizontal grid spacing, most of that cloudiness is on the sub-grid scale. Within 750 km of the cold front, the surface fluxes are a maximum (300 W m$^{-2}$ latent and 100 W m$^{-2}$ sensible). The PBL is as deep as $\sim$1.5 km and becomes slightly more decoupled. If present, additional dry air entrained into the top of the PBL via the dry intrusion mechanism could be aiding in increasing the turbulence to deepen and decouple the PBL in this region with decreased subsidence and weaker inversions, as suggested by Ilotoviz et al. (2021) and Tornow et al. (2023). Though our results are consistent with the presence of a dry intrusion, without an extensive back trajectory analysis, we cannot definitively say that the cyclone studied is influenced by dry intrusions and could just be regular post frontal conditions. Closer to the front, in an environment reminiscent of where the stratocumulus transition to cumulus in the subtropics, the cloud cover decreases to about 0.6 and the LWP increases to 100 g m$^{-2}$. The frontal sector is characterized by upward vertical motion, resulting in forced convective clouds with high LWP (> 100 g m$^{-2}$). The inversion strengths (5 K) and surface fluxes (100 W m$^{-2}$ latent and 0 W m$^{-2}$ sensible) are much weaker than the cold sector. The warm sector has the weakest surface fluxes (50 W m$^{-2}$ latent and 0 W m$^{-2}$ sensible), owing to the shallowest (< 500 m) and most stable PBLs with the lowest cloud cover (< 0.2) and LWPs (< 20 g m$^{-2}$). Weak vertical motion results in weak inversion strengths (< 1 K).

Despite representing the large-scale meteorology well, the COAMPS thermodynamic profiles during the northerly wind period at ENA are slightly too cool and moist, which results in clouds with a lower cloud base and larger thickness compared to observations. The weaker inversion strengths in COAMPS (1 K less than soundings) lead to more diffuse cloud boundaries at the inversion. The free tropospheric air entrained into the cloud layer is much cooler and drier compared to observations, which prevents the COAMPS profiles from becoming decoupled, and could explain the lower COAMPS cloud bases, which are near the lifting condensation level (LCL). In contrast, the soundings have cloud bases 200-300 m above the LCL due to the warmer air entrained into the cloud layer, leading to a decoupling of the cloud and subcloud layers. PBL depths are overestimated in COAMPS by $\sim$100 m, roughly the vertical grid spacing at that height. This result is different from most other COAMPS studies which find an underestimate in PBL depth by $\sim$200 m (Wang et al., 2011; Wyant et al., 2010, 2015). However, this prior COAMPS bias may be due in part to an unrepresentative performance of the Richardson number threshold in determining PBL depths (Eleuterio et al., 2004). We instead introduce a new thermodynamic gradient-based method based on the maximum of the second derivative of the liquid water potential temperature, which improves the PBL depth estimation.

We have tested the sensitivity of cloud and boundary-layer properties to different microphysics parameterization with varying

parameters. The replacement of Kessler (1969) warm rain microphysics with our KK Lite parameterization improves forecasts of precipitation and boundary-layer depth without adding any computational costs. Sensitivity tests varying the microphysical

parameters in KK Lite does not further improve the forecasted cloud behavior but yields relationships between boundary-layer, cloud, and precipitation properties largely consistent with those found in observations. Cloud properties likely are more sensitive to the particulars of the boundary layer parameterization (Juliano et al., 2019). Lamraoui et al. (2019) found that the degree of decoupling is sensitive to the PBL scheme, but modeled cloud fraction is more sensitive to the treatment of shallow convection, which in our simulations is quite crude. However, evaluating the PBL and shallow convective scheme sensitivity

in COAMPS is left for a future study. All simulations overestimate the LWP and underestimate precipitation and subcloud evaporation. This result is opposite to other findings, which report an underestimation of boundary-layer cloud LWP and cloud thickness and overestimation of precipitation in subtropical domains (e.g., Nelson et al., 2016; Wang et al., 2011). We speculate the reason for this discrepancy is related to the important difference between these low clouds in the post-cold-frontal region and subtropical stratocumulus whereby the strong surface fluxes associated with substantial grid-point vertical motions are less

present in marine subtropical environments where surface fluxes are weaker (cf. Ackerman et al., 2009; Stevens et al., 2005).

In this sense, the post-frontal low clouds have a strong element of similarity with clouds associated with cold-air outbreaks. In COAMPS, the large surface fluxes appear to be responsible for driving resolved vertical motions and associated clouds. Mechem and Kogan (2003) argued that the grid-scale cloud and vertical velocity features at mesoscale-model grid spacings

represented cloud ensembles and the emergence of mesoscale organization. Specifically, we interpret the cloudy cells that COAMPS produces as mesoscale organization of the broken cumulus that were observed by the KAZR after the cold front passage and are typically found in the wake of midlatitude cyclones (Field and Wood, 2007; Naud et al., 2018a) and in cold air outbreaks (Geerts et al., 2022; McCoy et al., 2017). Resolved COAMPS cloud fraction is underestimated compared to satellite, but the sub-grid contribution is overestimated. These inaccurate predictions of cloud cover could lead to large model biases

in albedo and incoming shortwave radiation (McCoy et al. 2017). Therefore, much work is needed to improve the representation of boundary-layer clouds within midlatitude cyclones. We expect that forecasts may improve with improvements to the boundary layer parameterization (Field et al., 2014; Zheng et al., 2024) and horizontal resolution, so as to not overrepresent grid-scale convection associated with instability driven by large surface fluxes. In addition, the balance between large surface fluxes and subsidence needs to be better represented so that the inversion strengths and entrainment into the cloud layer are

better predicted.

*Code and data availability.*  Analysis codes are available at https://github.com/jeissner/COAMPS-transect-analysis. Questions about the software can be addressed to the corresponding author. All ARM data sets from ENA used in this study are available through the ARM discovery website: https://adc.arm.gov/discovery/#/results/site_code::ena. ERA5 reanalysis can be downloaded from the ECMWF Climate Data Store: https://cds.climate.copernicus.eu/datasets.

*Author contributions.* JME and DBM jointly conceived the experimental design and writing of the paper. JME conducted the simulations, conducted the analysis of both model output and observations, and made the figures. YJ assisted with the model, providing data for model ingest, contributed to experimental design, and contributed to writing the manuscript. VPG contributed observational retrieval products, scientific interpretation, and contributed with writing the manuscript. JFB provided invaluable contributions to the automated synoptic ID component of the study.

*Competing interests.* The authors declare that they have no conflict of interest.

*Acknowledgements.* We acknowledge the insightful comments from several anonymous reviewers, which helped us better articulate the motivation for this work and ground it to the literature on subtropical boundary-layer clouds. Authors Eissner and Mechem were supported by ONR grant N00014-20-1-2519. We also acknowledge partial support from U.S. Department of Energy Atmospheric Systems Research Grants DE-SC0016522 and DE–SC0023083. This research is partially supported by the Office of Naval Research program element 0602435N
(author Jin). Author Ghate was supported by the U.S. Department of Energy's (DOE) Atmospheric System Research (ASR), an Office of Science, Office of Biological and Environmental Research (BER) program under Contract DE-AC02-06CH11357 awarded to the Argonne National Laboratory. Author Booth was partially funded by NOAA Award NA19OAR4310076. This work was performed at the HPC facilities operated by the Center for Research Computing at the University of Kansas supported in part through the National Science Foundation MRI Award #2117449.

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

**Table 1.** List of COAMPS sensitivity simulations run. Each sensitivity simulation is a variation of the control simulation, with the modifications noted.

| Microphysics parameterization | $N_r$, $N_c$ constants | Number of vertical grid points | Purpose |
|---|---|---|---|
| KK Lite | $N_r$=0.01, $N_c$=100 | 45 | control |
| Kessler | | 45 | microphysics parameterization sensitivity |
| Thompson | | 45 | microphysics parameterization sensitivity |
| KK Lite | $N_r$=0.01, $N_c$=40 | 45 | $N_c$ sensitivity |
| KK Lite | $N_r$=0.1, $N_c$=100 | 45 | $N_r$ sensitivity |
| KK Lite | $N_r$=0.1, $N_c$=40 | 45 | $N_r$, $N_c$ mutual interaction sensitivity |
| KK Lite | $N_r$=0.1, $N_c$=100 | 87 | vertical grid sensitivity |

**Table 2.** Cloud-conditioned ENA means (top values) and fine mesh domain means (edge points of fine mesh removed) (bottom values in italics) for each microphysics parameterization and observations during the northerly wind period of the case study.

| | Observed | Kessler | Thompson | KK Lite | KK Lite $N_c$=40, $N_r$=0.01 | KK Lite $N_c$=100, $N_r$=0.1 | KK Lite $N_c$=40, $N_r$=0.1 |
|---|---|---|---|---|---|---|---|
| Cloud base RR (mm d$^{-1}$) | 6.06 | 1.63 *2.98* | 1.45 *1.72* | 1.82 *8.18* | 2.65 *8.36* | 0.40 *2.65* | 0.67 *2.85* |
| Surface RR (mm d$^{-1}$) | 0.87 | 0.75 *2.23* | 0.27 *1.04* | 0.61 *6.14* | 1.06 *6.12* | 0.02 *1.56* | 0.09 *1.62* |
| Subcloud evaporation rate (mm d$^{-1}$) | 5.17 | 0.88 *0.75* | 1.17 *0.67* | 1.21 *2.04* | 1.59 *2.24* | 0.38 *1.09* | 0.58 *1.22* |
| PBL depth (m) | 1683.2 | 1529.2 *1860.7* | 1752.0 *2114.7* | 1715.5 *1978.3* | 1619.3 *1877.5* | 1619.0 *1957.4* | 1581.9 *1961.5* |
| LWP (g m$^{-2}$) | 55.3 | 54.2 *39.6* | 127.3 *92.4* | 113.3 *97.4* | 119.0 *92.8* | 201.6 *133.7* | 191.8 *127.6* |
| Cloud thickness (m) | 294.7 | 351.5 *336.9* | 500.1 *538.4* | 434.5 *446.9* | 442.8 *434.1* | 521.3 *496.5* | 517.7 *483.4* |
| Low cloud cover (%) | 76.1 | 84.6 *74.8* | 87.4 *74.1* | 81.5 *71.8* | 82.9 *72.6* | 85.5 *73.9* | 81.6 *74.4* |

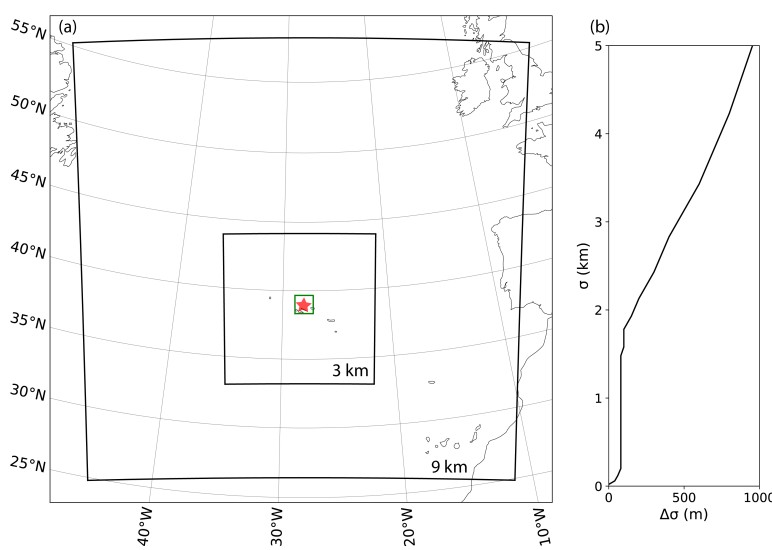

**Figure 1.** (a) COAMPS doubly nested domain centered on Graciosa Island (red star). The 144 km × 144 km analysis box is outlined in green. (b) COAMPS vertical grid spacing ($\Delta\sigma$) with height ($\sigma$) over the lowest 5 km.

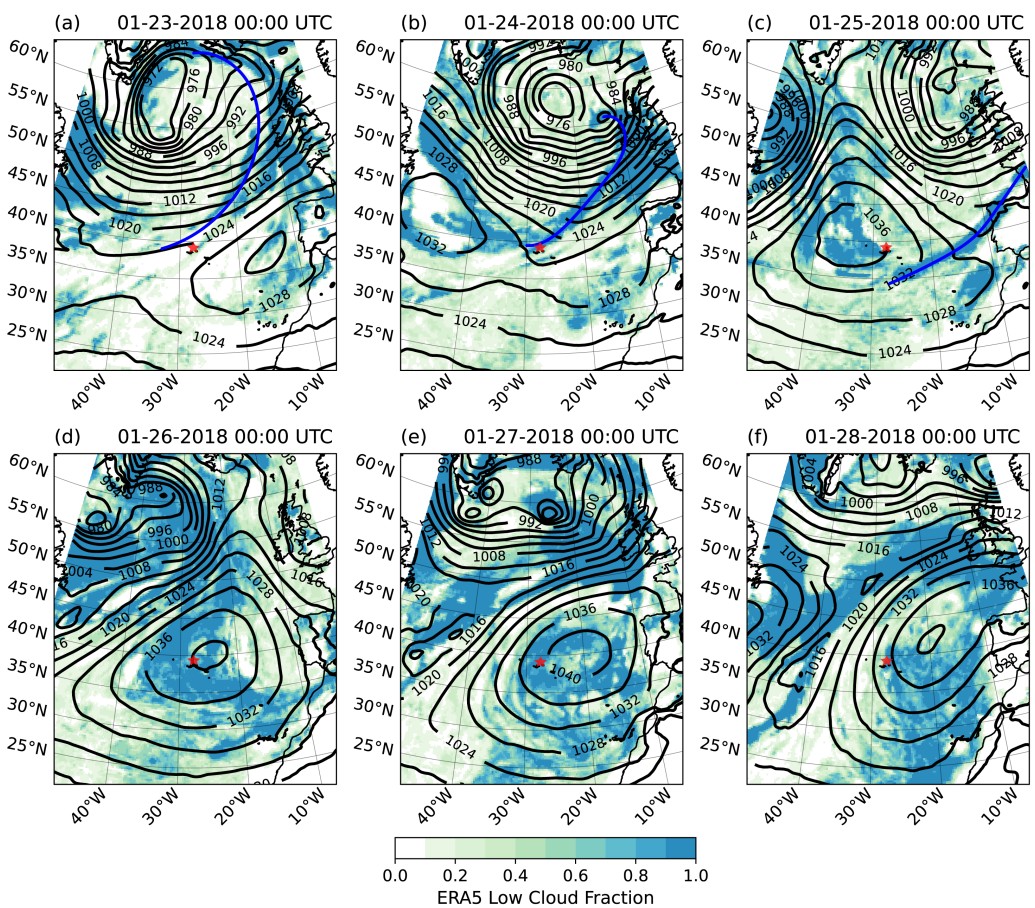

**Figure 2.** ERA5 SLP (black contours), low cloud fraction (color-filled contours) in the north Atlantic at 00:00 UTC 23-28 January 2018. Graciosa Island is marked by the red star, and the cold front is shown by the blue line.

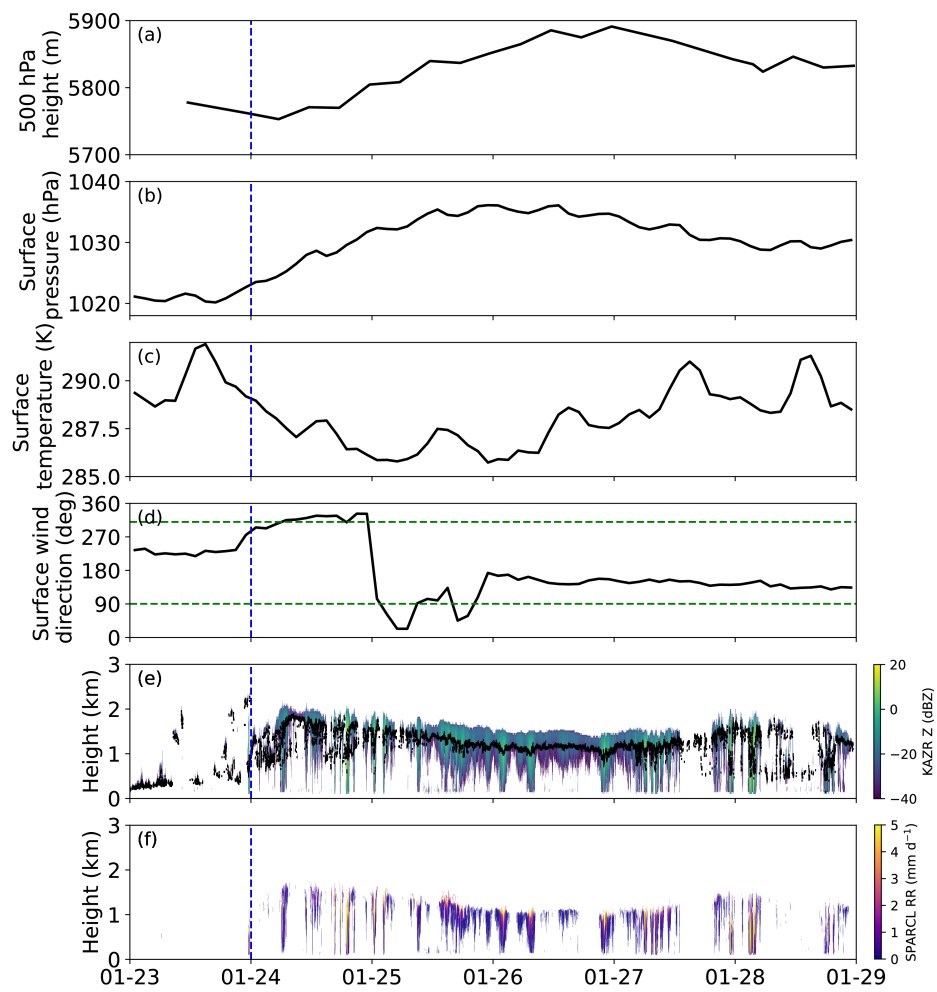

**Figure 3.** ENA observations over the simulation period (23-28 January 2018). (a) 500 hPa heights from the soundings, (b) MET surface pressure, (c) MET surface temperature, (d) MET surface wind direction, with green horizontal lines indicating the bounds between northerly and southerly winds, (e) KAZR radar reflectivity and ceilometer cloud base height, denoted by the black dots, (f) SPARCL subcloud drizzle rates. The passage of the cold front is shown by the vertical blue line.

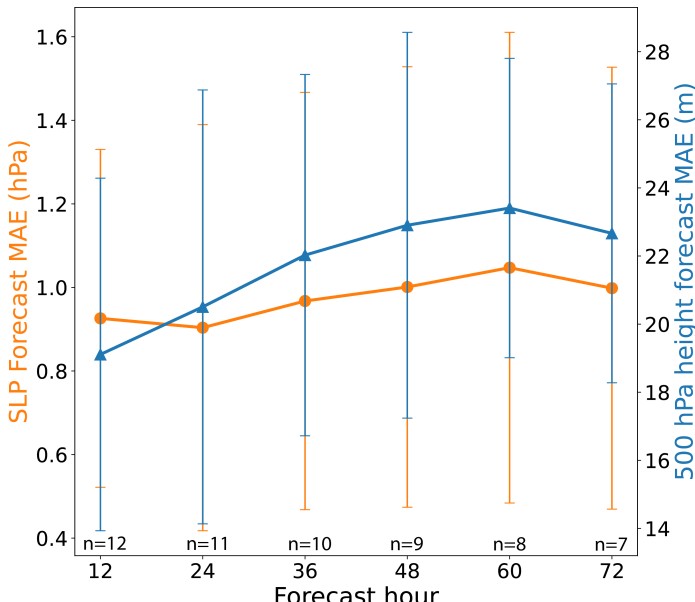

**Figure 4.** SLP (orange) and 500 hPa geopotential height (blue) mean absolute error (compared to ERA5) versus the forecast hour for a series of simulations. The error bars indicate the standard deviations between each simulation for each forecast hour. The number of simulations (n) included in the mean absolute error calculation for each forecast hour is listed.

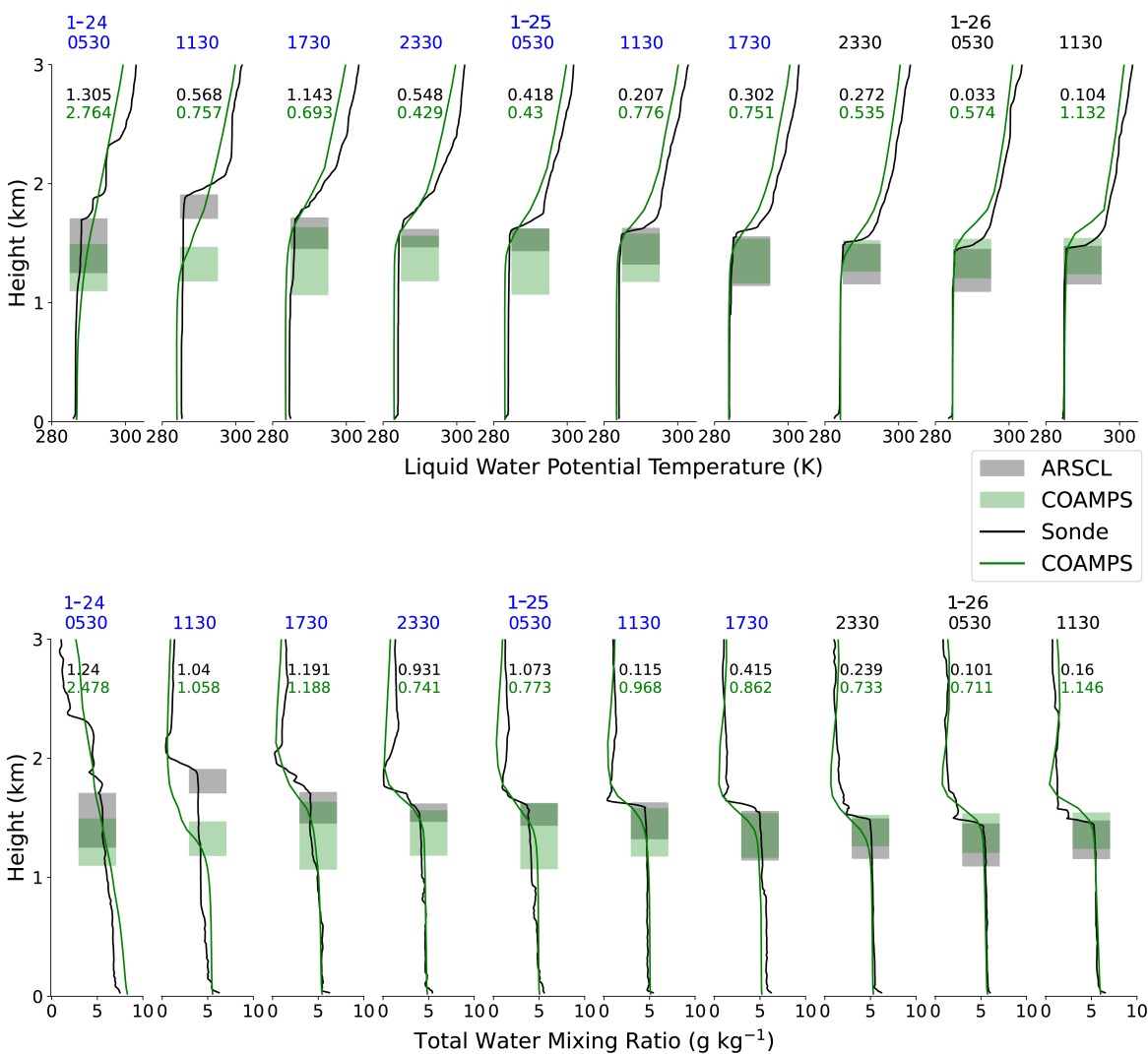

**Figure 5.** Liquid water potential temperature (top row) and total water mixing ratio (bottom row) profiles from sounding (black) and the COAMPS control simulation (green) throughout the study period. ARSCL cloud boundaries are colored light grey, COAMPS cloud boundaries are colored green, and the overlap between the two is dark green. The blue dates indicate the soundings taken during the northerly wind period and the black dates are from the southerly wind period. Numbers at the top of each plot indicate the decoupling index calculated from the sounding (black) and COAMPS (green) liquid water potential temperature (top row) and total water mixing ratio (bottom) profiles.

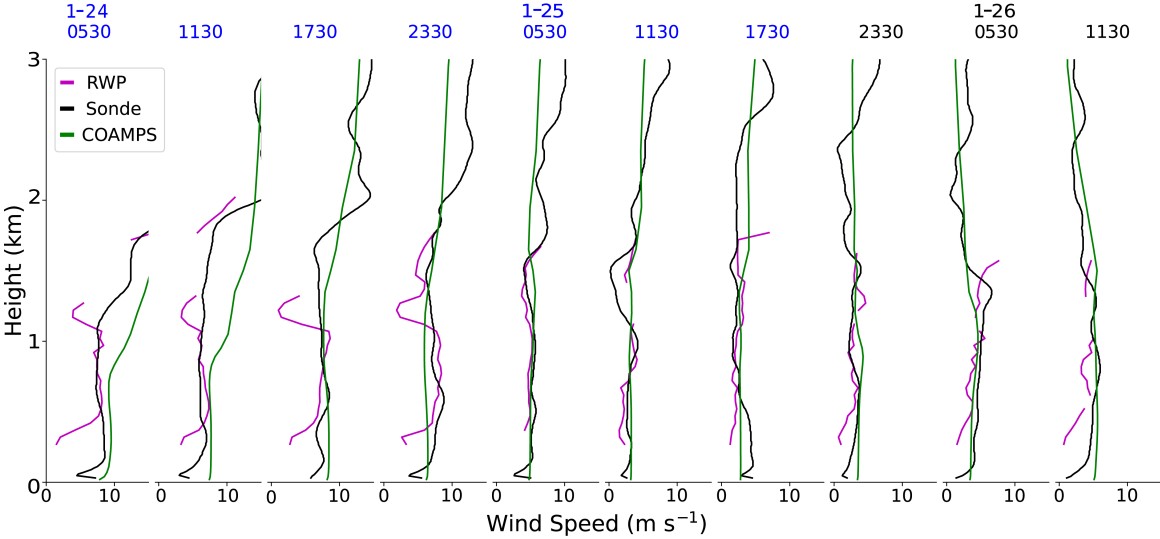

**Figure 6.** Same as Fig. 5 but for profiles of Radar Wind Profiler, sounding, and COAMPS wind speeds.

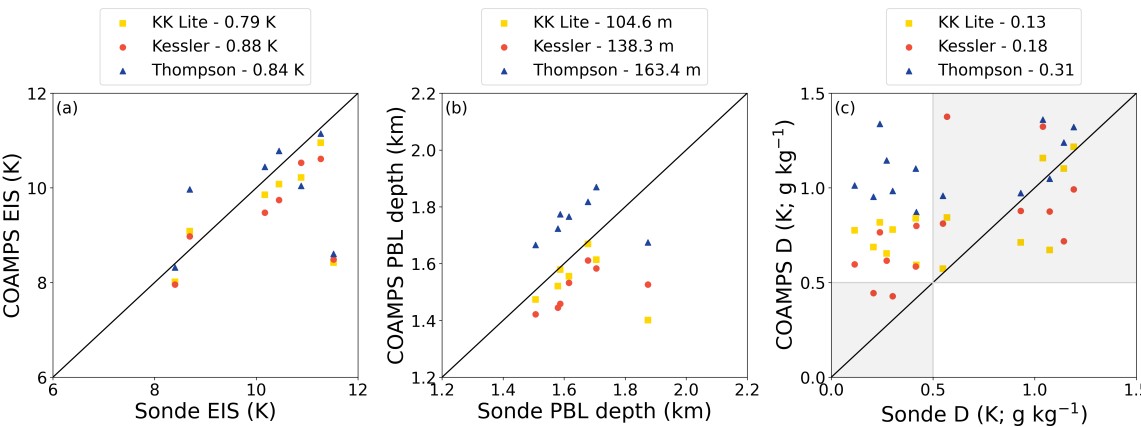

**Figure 7.** COAMPS vs sounding (a) inversion strength (EIS), (b) PBL depth, and (c) decoupling parameters calculated from both the liquid water potential temperature and total water mixing ration profiles during the northerly wind period of the case study (7 total soundings). The 1:1 line is plotted in black, and the mean absolute error between the observed and each COAMPS microphysics scheme is listed in the legend.

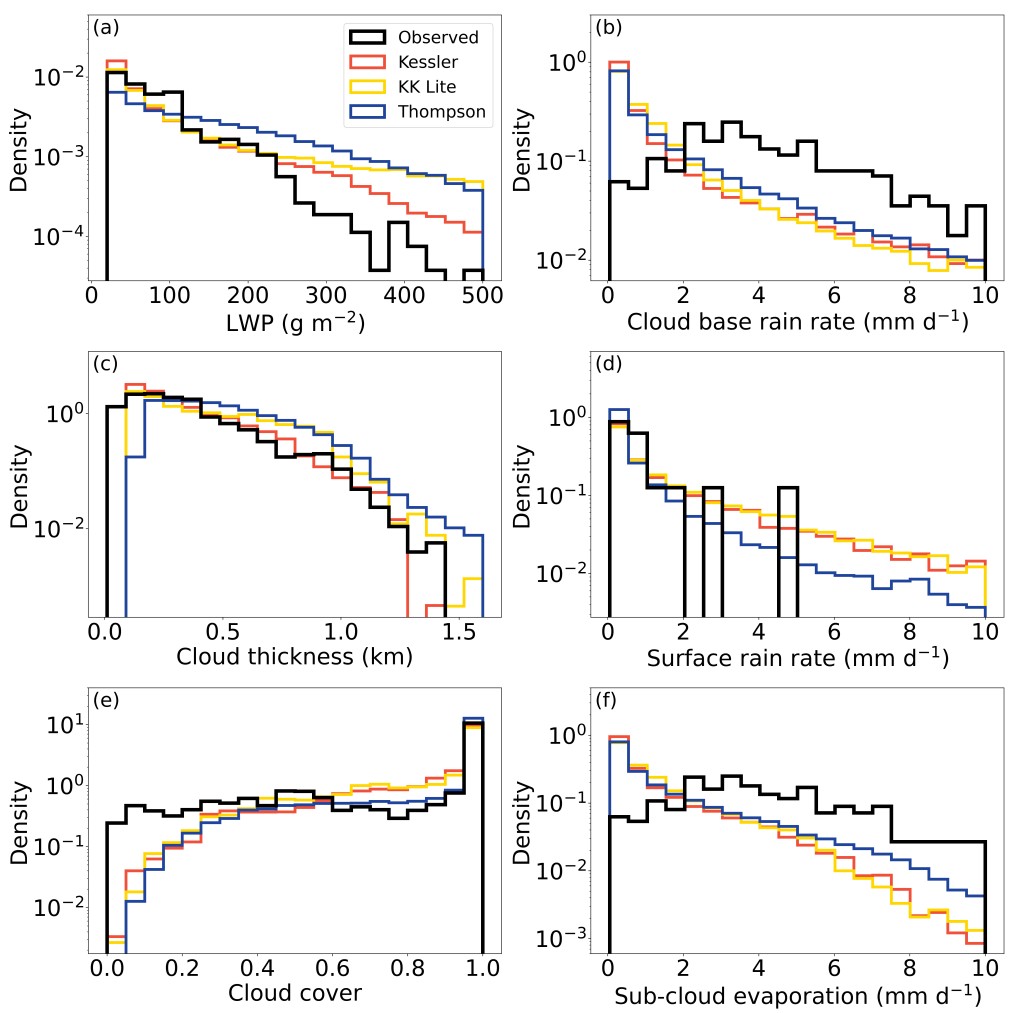

**Figure 8.** PDF of observations (black) and each microphysics scheme during the northerly wind period of the case study.

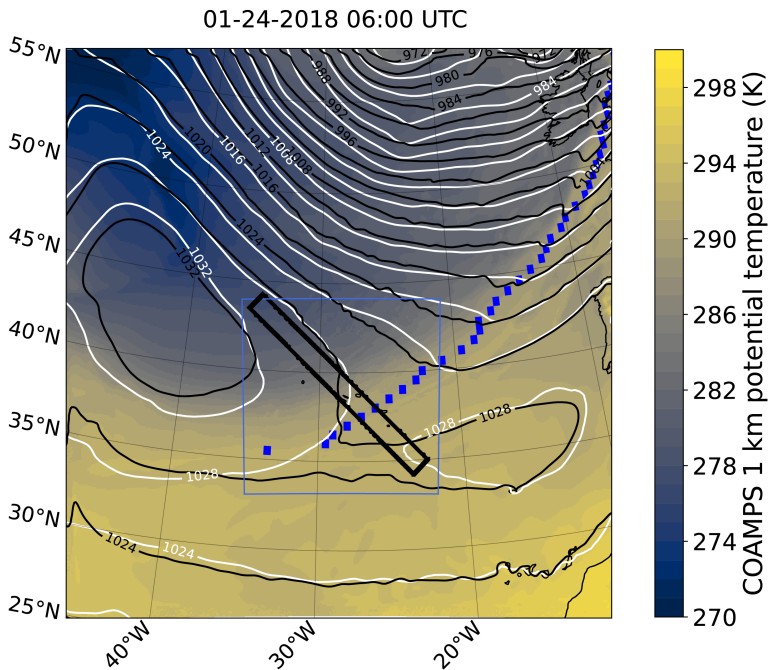

**Figure 9.** COAMPS coarse mesh SLP (black contours) and 1 km potential temperature (color-filled contours), ERA5 SLP (white contours), and identified cold front using COAMPS data (blue dots). The outline of the fine mesh is the blue box, and the transect width is outlined by the thick black lines.

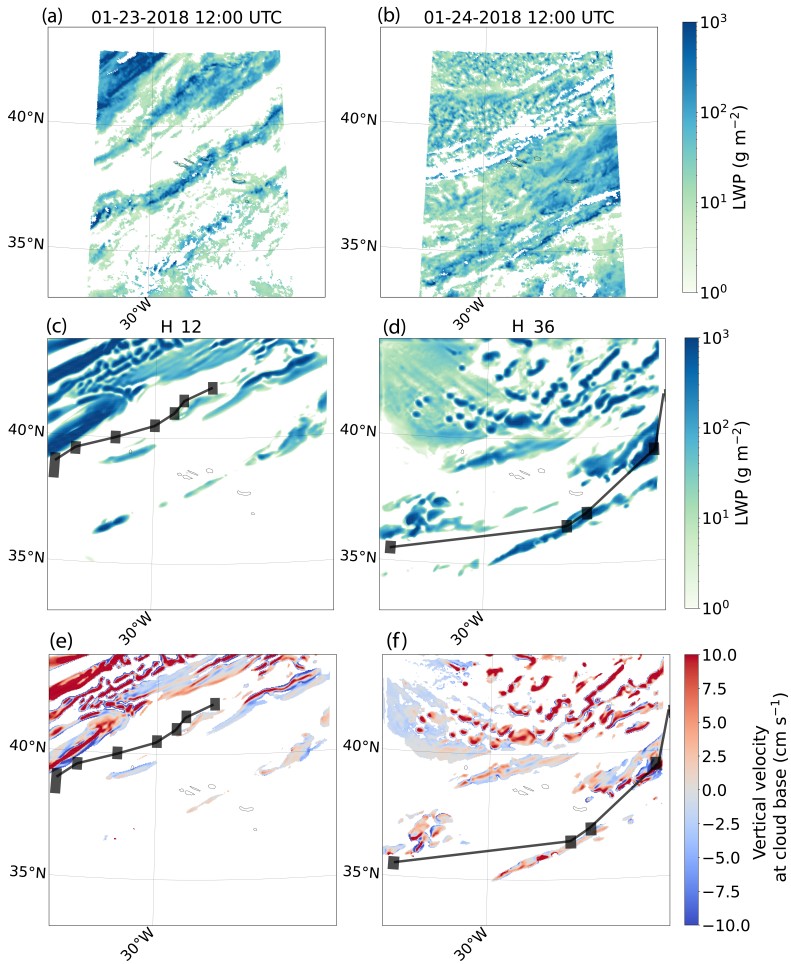

**Figure 10.** Satellite-derived LWP (top row), COAMPS LWP (middle row), and COAMPS cloud base vertical velocity (bottom row) for 12:00 UTC on 23 January (left column) and 24 January (right column) and the equivalent COAMPS forecast time. The location of the algorithm-identified cold front is shown by the black squares and lines.

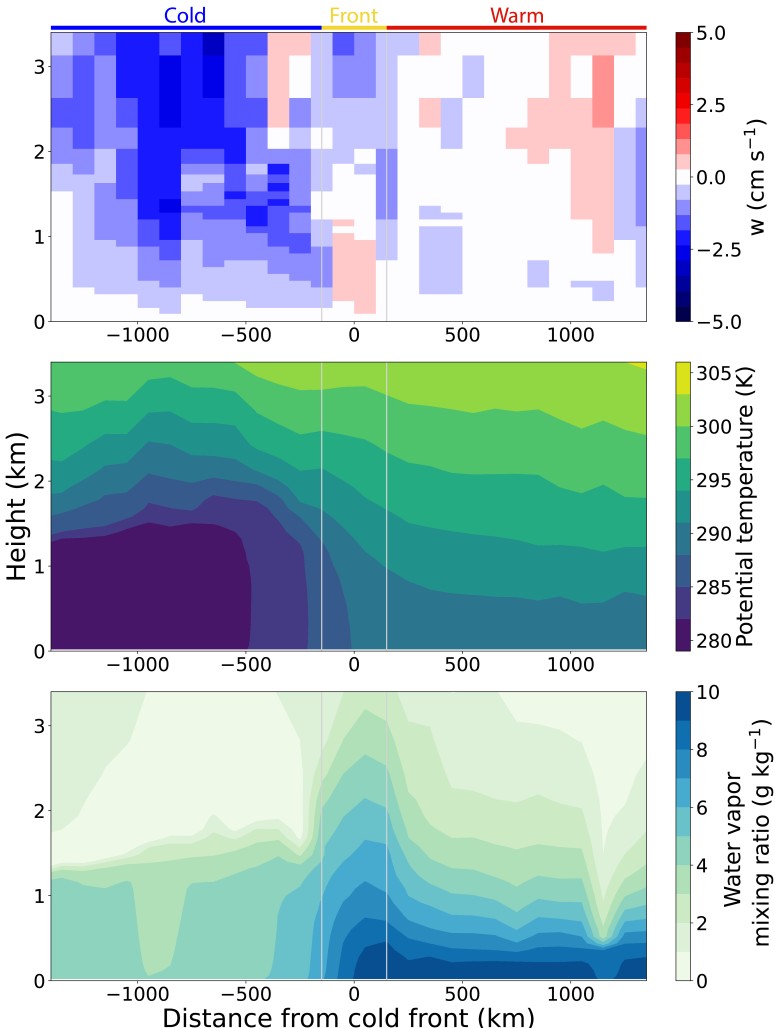

**Figure 11.** Vertical cross sections along the transect of the median in 100 km bins of 8 model forecast times of (top) vertical motion, (middle) potential temperature, and (bottom) water vapor mixing ratio. The 0 distance marks the location of the cold front. We chose to show the median of the 8 transects but using the mean does not change the results.

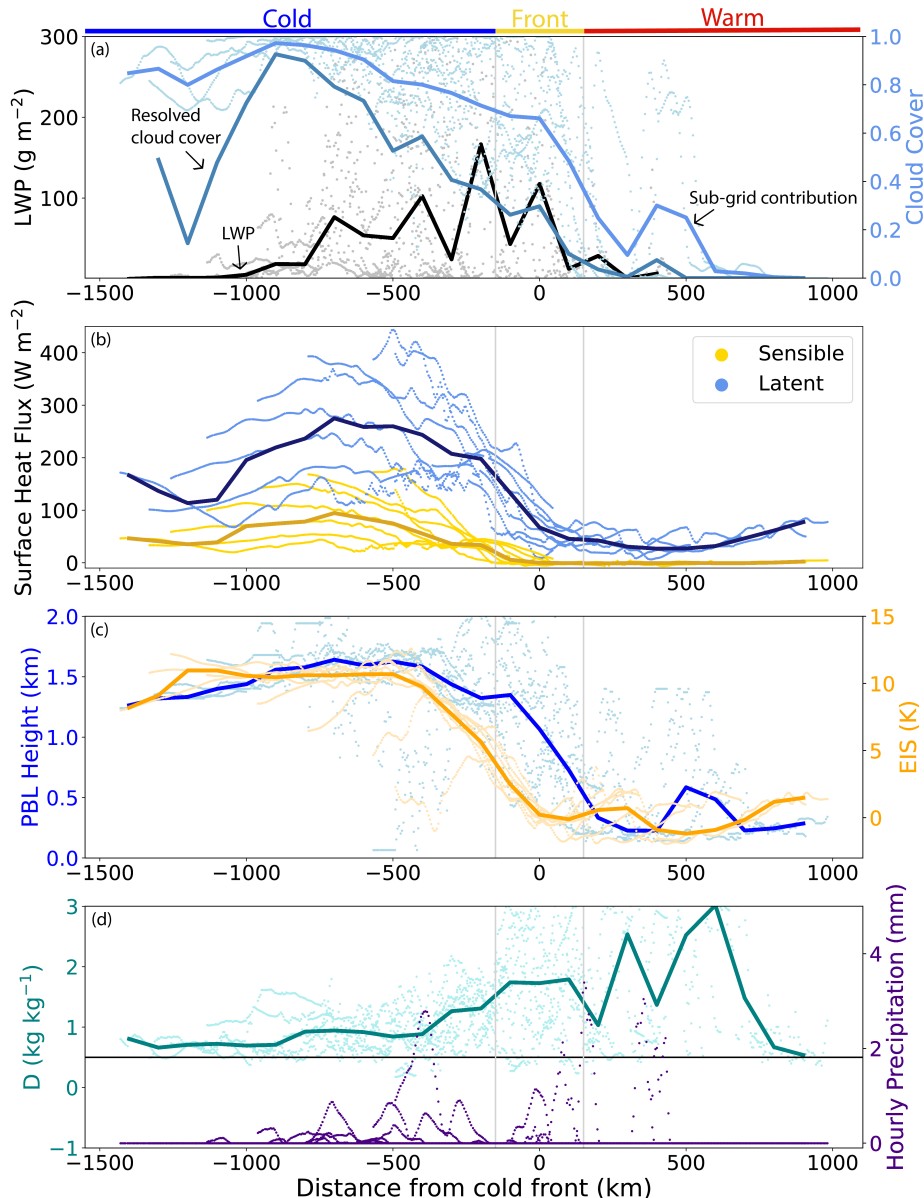

**Figure 12.** Cold-front-centered transects for 8 different forecast hours (forecast hours 12, 18, 24, 30, 36, 42, 48, and 54). Thin lines or dots represent each transect at a given time and the thicker lines represent the median in 100 km bins of all composites. LWP is conditioned on non-zero values. The decoupling indices in panel (d) are calculated using the total water mixing ratio.

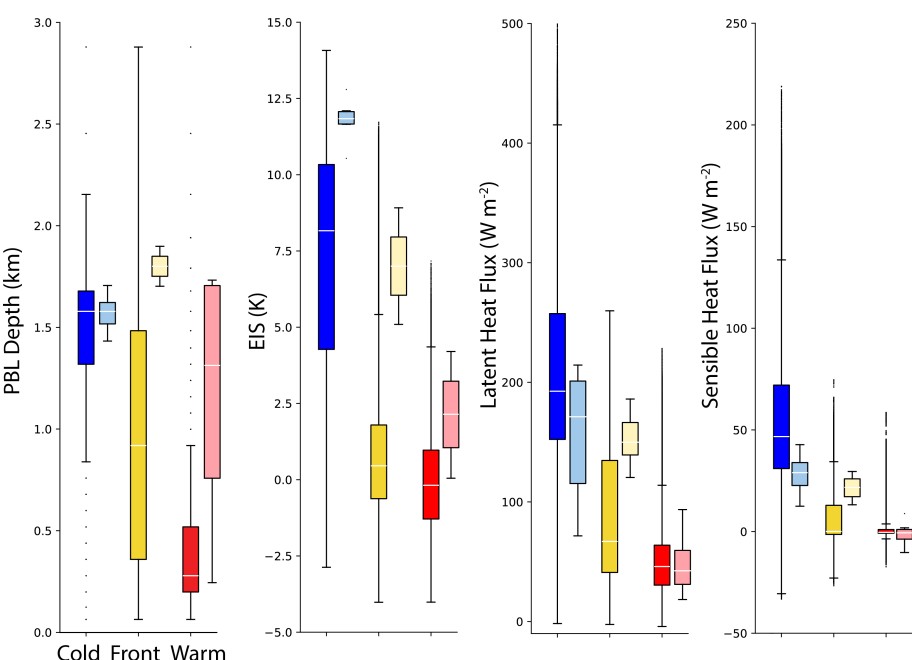

**Figure 13.** Box plots of COAMPS environmental variables (darker colors) and observations (lighter colors) in each the cold (blue), frontal (yellow), and warm (red) sectors.

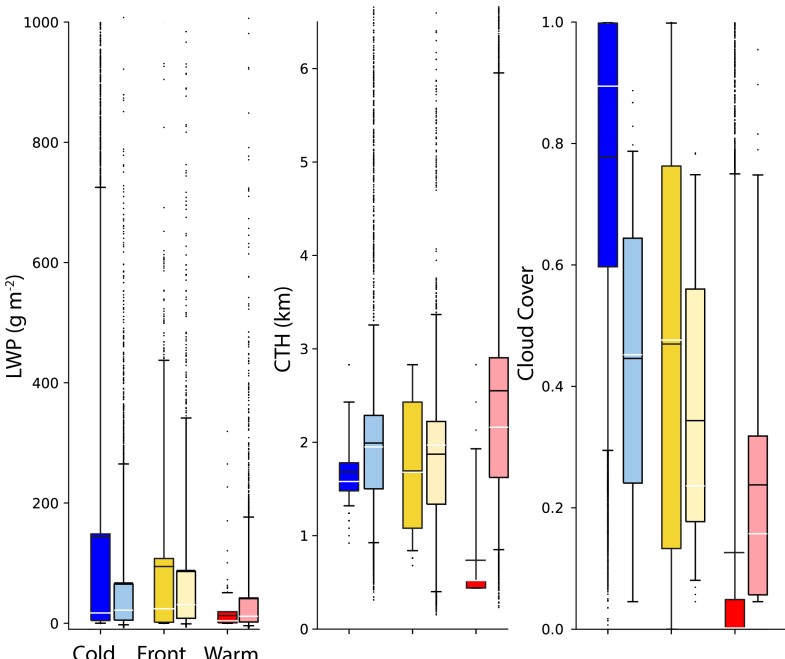

**Figure 14.** Box plots of COAMPS cloud variables (darker colors) and satellite observations (lighter colors) in each the cold (blue), frontal (yellow), and warm (red) sectors. Median values are shown by the white line, and mean values are the black line. LWP is conditioned on non-zero values and COAMPS cloud cover includes the sub-grid contribution.

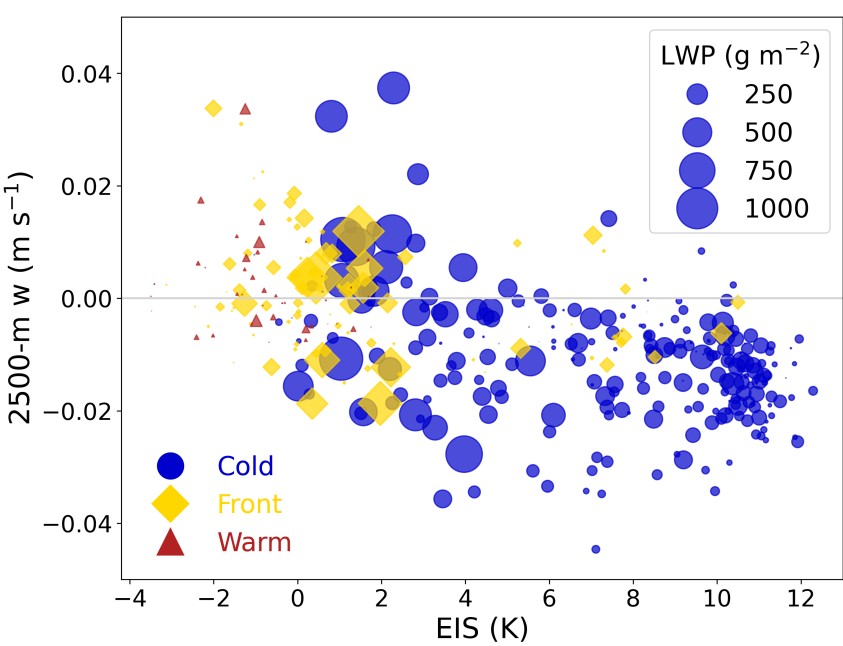

**Figure 15.** 2500 m vertical velocity versus EIS, color-coded and shaped by the sector: cold (blue circles), warm (red triangles), and frontal (yellow diamonds). The size of the marker indicates the LWP for all shapes. Each marker represents the average of 144 × 144 km boxes throughout the fine mesh at 8 different forecast hours.