# Peer review of "Identifying Synoptic Controls on Boundary Layer Thermodynamic and Cloud Properties in a Regional Forecast Model"

_EGUsphere, 2024_

## Referee Comment (RC1)

Manuscript evaluation of "Identifying Synoptic Controls on Boundary Layer Thermodynamic and Cloud Properties in a Regional Forecast Model"

**General comments**

This paper investigates how marine boundary layer clouds respond to synoptic conditions, particularly in the cold sector of a cyclone, and evaluates how a numerical weather prediction model compares to observational data. This is an important and timely research topic, given the ongoing challenges in accurately representing these clouds in models and their critical role in Earth's radiative budget. The study's use of a novel approach—transforming coordinates into a frontal perspective (following Naud et al. 2016)—is a valuable contribution, particularly for the modeling community, as it allows for improved comparisons between simulations and observations.

However, the manuscript has some weaknesses that should be addressed:

- **Clarity and Contextualization:** The introduction lacks clarity in some areas and should define the goals of the paper or contextualize it within existing research more clearly (see SC1 and SC2). The overarching goal of the study is not explicitly stated early on, making it harder to follow the intended contribution. Additionally, the authors should add a stronger connection to previous work, explicitly stating what knowledge gap this study addresses.

- **Interpretation of Figures & Takeaways:** Some figures contain interpretations that are incomplete or potentially incorrect (SC3). Additionally, discussions of figures often lack clear takeaway points (SC5), making it difficult for the reader to track key findings throughout the manuscript. This also affects the conclusion, as it becomes harder to recall where certain results were discussed.

- **PBL Depth Discrepancy:** There is an inconsistency when reporting differences in planetary boundary layer (PBL) depth using two different methods (SC4), and the implications of this discrepancy for commonly used approaches in the field are not discussed.

- **Framing and Significance of Findings:** One aspect not covered in the specific comments is that the authors may be underselling the importance of their results. For example, the discrepancy between open-cellular convection observed in satellite imagery and closed-cellular convection in the model is potentially significant, given its implications for cloud albedo, climate feedbacks, and solar radiation management. The authors should consider whether these aspects deserve more discussion, either through an expanded interpretation of the findings or by referencing relevant literature.

**Specific comments**

**SC1)** Adding clarity in the Introduction

The second paragraph of the introduction (lines 24–36) is difficult to follow and would benefit from revision to improve clarity. For instance, the sentence *"this tends to decouple the boundary layer, where mixing between the cloud and surface layers is inhibited"* (line 30) is ambiguous. It raises two questions:

1. What exactly does the boundary layer decouple from?
2. Does the sentence imply that mixing is inhibited because the boundary layer is decoupled, or is the decoupling a result of inhibited mixing?

I suggest rewriting this paragraph to provide clearer explanations of the mechanisms involved and how they interact.
* * *
**SC2)** Make clear the goal of the paper and better contextualize it within current research (Introduction)

The introduction explains the mechanisms relevant to marine low-level clouds and the transitions between different regimes, which are essential points for the reader to understand. However, two critical elements are missing:

1. **The goal of the paper**: It is unclear what the overarching goal of the paper is (is it a proof of concept, are we looking for behaviour that has not yet been analyzed?). For instance, the last paragraph in the introduction alludes to an investigation of multiple "systems" (plural) but as far as I understand it, only a single case study is investigated. The authors should clearly state the aim of the paper upfront to provide readers with a cohesive understanding of the study's purpose and structure. Also consider adding a short sentence regarding these points in the abstract.

2. **Contextualization within current research**: The introduction does not adequately explain what is novel about this study or how it addresses a specific gap in knowledge. For instance:

   ● What gap in knowledge is being addressed exactly?
   ● What is the primary motivation for evaluating the regional numerical weather prediction model in this context?
   ● Are midlatitude baroclinic synoptic systems especially important or poorly understood? This last point should be addressed in the paragraph in line 38 to 56.
* * *
**SC3)** Incorrect/incomplete interpretation of figures

I have the following issues with the analysis of results presented in Figures 5 and 7:

1. Line 285: COAMPS only underestimates the temperature profile throughout the lowest 3 km of the atmosphere from 11:30 on 24.01 to 11:30 on 25.01, not for the time frame stated by the authors.
2. The line immediately following: COAMPS only overestimates PBL moisture for the first two subplots of Fig. 5 (24.01 05:30 to 24.01 11:30) and not as stated by the authors.
3. Directly thereafter: to my eyes, the Thompson param. and Kessler param. show similar skill in predicting the decoupling (Fig7c). Please add some sort of metric to support the statement that one is better than the other in this regard.
4. Line 307: not all COAMPS microphysics sensitivity experiments overestimate stability, because unless I have understood something, there are three data points below the 1:1 line in Fig 7 a).
5. Line 315: "The Kessler parameterization [...] only [overestimates] decoupling for **one** sounding" → how are you defining an over/underestimation? Because there is definitely **more** than one Kessler data point above the 1:1 line in Fig. 7c).

———————————————————

**SC4)** Richardson number vs. best estimate approach

In line 130 the authors state that the best estimate approach results in PBL depths that are **800m** larger than those derived from the Richardson number approach. In line 219 the authors put this number at **400m**. Please revise (or in case this is not a mistake, make clear why the difference is 800m and then 400m).

Please also discuss the implications this discrepancy has for methods commonly used in the field.

———————————————————

**SC5)** Emphasize main findings

In the paragraph between lines 398 and 409, as well as the following paragraph discussing Figures 11 and 12a, the authors provide a brief summary of what is shown in the figures. However, the key takeaways are only stated briefly.

Clarity would improve if the authors more clearly emphasize the main findings—e.g., by explicitly stating, *'As discussed in Section X, this shows that the model is incapable/capable of [etc.]'* or similar phrasing—to guide the reader and reinforce the significance of the results.

For instance, in Line 415–416 (*'Subsidence dominates [the entire cold sector]'*), could you briefly remind the reader whether this behavior aligns with expectations? The first paragraph in page 14 is a positive example and does a good job at this.

While most of these takeaways should, of course, be reserved for the conclusion, these modifications would still improve readability. These modifications would also help the reader recall where specific findings were discussed when reaching the conclusion.

———————————————————

**SC6)** Use color deficient friendly color palettes and (perceptually uniform) colormaps

Figures 12b, 13, 14 and 15 use red and green colors on the same plot. This combination is not color-deficient friendly, please a different color palette (see e.g https://www.nceas.ucsb.edu/sites/default/files/2022-06/Colorblind%20Safe%20Color%20Schemes.pdf for a sensible choice of colors)

Figure 9 uses a rainbow colormap, which is neither color-deficient friendly nor perceptually uniform. Please change the colormap (viridis, plasma, batlow, etc.).

**Technical corrections**

- Please add relevant citations in line(s) 18, 20-22, 25.
- Line 309: make sure you are referencing the correct figure. Also in the rest of the paragraph.
- Figure 10: the labels "a) b), … i)" are missing
- Use either "Fig. X" or "Figure X"

---

## Referee Comment (RC2)

Review **"Identifying Synoptic Controls on Boundary Layer Thermodynamic and Cloud Properties in a Regional Forecast Model"** by Jordan Eissner et al.

This is an interesting study which discusses the boundary layer cloud properties of a midlatitude synoptic cyclone. It uses regional weather model results, including sensitivity tests, as well a detailed comparison with various type of observations. Given the broad scope of the study, i.e. the study of boundary cloud (dynamics) in a large-scale environment, this is potentially interesting for both communities. Therefore, the framing is quite important, however here is still quite some improvement possible. I am reviewing this manuscript from a perspective on my expertise on large-scale atmospheric dynamics, and therefore this review has a slight focus on that.

1. I think the most important part which can be improved is the general motivation on what is new in this particular study and how it fits with earlier studies. There has been quite a lot research already on clouds and precipitation related to midlatitude cyclones. It would be good that the research gap is explicitly stated. In the conclusions the authors mention (lines 566-569) that they *"In this work, we demonstrate that the analysis method provides a pathway to compare boundary-layer cloudiness forecasts against observations in the context of the different sectors of midlatitude cyclones and in a manner that minimizes errors associated with misplacement of the cyclone center and frontal structures (i.e., phase error)."* If this is the main gap the study tries to address it would be good to explicitly state this in the introduction. Moreover, it would be good to also more explicitly state what 'the analysis method' refers too.

2. Related to this, in the first paragraph two distinct cloud-transition mechanisms are discussed: The deepening warming hypothesis and a cloud transition mechanism that relies on precipitation. Given the first objective of the study ("*to apply our understanding of boundary-layer clouds, largely relevant to barotropic atmospheres in the subtropics, and expand it to boundary-layer clouds accompanying midlatitude baroclinic synoptic systems"*, lines 80-81) some expectations are set that a direct comparison is made between these mechanisms and the case discussed. However, this is never explicitly done. I am not saying the authors should definitely do this, however I still find it hard to understand what the conclusion is regarding this first objective after reading the manuscript. This is also because in the conclusions the main focus is on the results regarding the second objective. It would be good to more explicitly state the conclusions regarding this objective.

3. The section on how the cyclones are detected and how the cross-frontal composites are made (section 5.1) is quite detailed, I would suggest to move that to the methods section, since that is also the section where the reader expects this information. Moreover, it is already used before in e.g. Figure 2.

4. The presentation of the figures is already much better with the corrections made, however could still be improved sometimes, see detailed comments below.

**Minor comments:**
Abstract line 1: It might be my ignorance, but can you really put this so strongly? Or do you refer mainly to the knowledge in a theoretical (boundary layer) framework? I think e.g. there is e.g. quite a lot of studies already e.g. on mesoscale rainbands already decades ago (e.g.

Houze et al., 1976, Matejka et al. 1981, Knight and Hobbs, 1988), and there are also quite some empirical studies on clouds associated with fronts and extratropical cyclones as well.

Line 8: "The Frontal region" does this refer to the cold or warm front or the fronts in general?

Line 41: Given the broad scope of the study and possible broad background of the readers, it might be good to introduce the regions of there one would large-scale vertical ascent in a typical extratropical cyclone.

Line 90-95: I think the discussion on the summer cloud properties in the Azores region is not strictly necessary and might be even be confusing, since this study focusses on a week in winter.

Line 100-104: I do not completely see why a positive NAO phase would not be representative for winter storm systems. I would earlier maybe state the opposite, since the negative phase is more associated with a blocked weather pattern over Europe. Furthermore, I would also argue that the position of the studied extratropical cyclone matches quite well with the average position of the storm tracks (see e.g. Neu et al., 2013). Therefore I would suggest to rewrite this part a bit.

Line 152: It might be good to already introduce the exact period of the model simulations, or remove this part completely here, since now it is scattered over two paragraphs.

Line 217: "Only over the ENA site" Which region is chosen for this?

Line 217-219: Given the use of the same Richardson constant of 0.5, do the authors have an idea why the underestimation in the regional model is only by about 400 meters, compared to the 800 meters using the sounding data?

Line 243-245: The system is already occluded probably, though it is hard to see from the chosen region, which might be the reason why your front is not originating at the centre. The phrasing suggests that this is not visible in the temperature fields, if this is not the case I would rewrite this part. Furthermore, as far I understand you are using a frontal detection at 1 km? Given the (potential) tilt of fronts with height, one would not always have 100 % correspondence anyways.

Lines 382-385/Figure 9: Why is the transect not centred in the middle of the cold front? And how is the exact location along the cold front determined?

Figure 2: Can the displayed region be extended a bit to the north? I understand that the main region of interest are the frontal regions, but now the studied cyclone is not (completely) visible at displayed time steps.

Caption Figure 2: "Red start" should be "Red star"

Figure 4: Over which region is the MAE calculated?

Figure 5: I also would suggest moving the legend out of the plotting area here. Moreover, the legend is also not completely clear to me, does it refer to the lines plotted or the coloured squares indicating the cloud boundaries?

Figure 8: The densities of panel (e) are beyond the y-axis. Futhermore I am a bit confused on that the total density (surface area below the lines) does not seem to add up to the same value for each subplot. Is this due to the different number of observations points in each of the subpanels?

Figure 10: I understand the choice of a blue colour to represent of the cold front, however given the blue colour map in most panels this makes it hard to distinguish from the background. I would suggest making the location of the front black.

Figure 11, caption: I wondered here why the authors to use the median here, since using only 8 forecasts might result in a skewed result. Have the authors compered to e.g. using the mean to see if the results would be very different? Moreover, is the median calculated at each point separately?

**References**:
Houze Jr, Robert A., et al. "Mesoscale rainbands in extratropical cyclones." *Monthly Weather Review* 104.7 (1976): 868-878

Knight, David J., and Peter V. Hobbs. "The mesoscale and microscale structure and organization of clouds and precipitation in midlatitude cyclones. Part XV: A numerical modeling study of frontogenesis and cold-frontal rainbands." *Journal of the atmospheric sciences* 45.6 (1988): 915-931.

Matejka, Thomas J., Robert A. Houze Jr, and Peter V. Hobbs. "Microphysics and dynamics of clouds associated with mesoscale rainbands in extratropical cyclones." *Quarterly Journal of the Royal Meteorological Society* 106.447 (1980): 29-56.

Neu, Urs, et al. "IMILAST: A community effort to intercompare extratropical cyclone detection and tracking algorithms." *Bulletin of the American Meteorological Society* 94.4 (2013): 529-547.

---

## Referee Comment (RC3)

Review of **"Identifying Synoptic Controls on Boundary Layer Thermodynamic and Cloud Properties in a Regional Forecast Model"** by Jordan Eissner, David Mechem, Yi Jin, Virendra Ghate, and James Booth

In this paper the authors examine the PBL and the low-level cloud structure in the different areas of a midlatitude cyclone. Using the COAMPS regional model they simulate the transition of a front over the ENA site on Gracioca island using the observations to validate the model results. The results emphasize the differences in the PBL structure between the different sectors of the cyclonic system.

I think that the subject of the study can be of great interest to a broad range of readers and the results may apply both to regional/cloud modelers and researchers in the large-scale dynamics field. However, in order to be ready for publication the manuscript needs to be restructured to help readers reach the main conclusions of the paper. The manuscript in some places repeats itself or presents information that does not serve the analysis. For example, the main objective of the paper is understanding the low-level cloud structure in baroclinic systems, with an introduction which is focused on cloud formation and transitions between cloud regimes and an extensive discussion on the different microphysical schemes but not enough information regarding past research on cloud structures in midlatitude cyclones. I suggest that the main objectives of the research and final conclusions and abstract be rewritten and clarified, and that the introduction adjusted to better serve the reader when presented with the conclusions. Lastly, I would consider rearranging the conclusion section to better follow the outline of the paper and to ease the reader into the final conclusions. Here I detail further major concerns and more specific issues.

**Major comments:**

1. Aims of the current study (lines 80-86): the aims as written in this paragraph do not reflect what is later achieved. Aim #1 does not apply knowledge on midlatitude systems, but only to one case study. Aim #2 is a methodological plan, but it is unclear what is the more specific scientific objective it aims to address? (e.g., examine parameterizations, assess operational NWP skill? This is unclear).

2. Missing consideration of cyclone airstreams in the baroclinic environment: The introduction and discussion should acknowledge the importance of cyclone-related airstreams for shaping the cyclone environment, including the PBL regimes and clouds. Recent papers address these issues. Specifically for ENA, Ilotoviz et al. (2021) highlighted the role of dry intrusions for shaping cold-sector PBL regimes and clouds. Turnow et al. (2023) note the importance of rain onset for cloud transitions using LES, explaining differences among cold sector regions closer Vs. away from the cyclone center. Liu et al. (2021) studied cloud transitions in post-cold fronts regions of Mediterranean cyclones, relating the transitions to precipitation dynamics. Vast literature relates warm conveyor

belts to clouds along the cold front and in the warm sector, while other studies relate surface fluxes to the passage of cyclones and/or cold fronts. Since one of the aims of the current study is to examine whether known PBL regimes apply to baroclinic environments, these aspects should be better introduced and discussed again in this context.

3. Overall, the representation of boundary-layer clouds in COAMPS is not good (Fig. 10 mainly, but also in terms of cloud cover etc.). Therefore, the notion in lines 506-507 should be toned down. I was left wondering what can we still learn from the results of the evaluation study. For example, one of the key finding is the relatively small influence of the microphysics scheme on cloud parameters (which somewhat raises the question why is this aspect so prominent in the methods section?). Given the model deficiencies, what do we learn more specifically that can guide future model developments?

4. In the introduction (Lines 24-36) there is a long discussion on the transition mechanisms between the different cloud fields regimes. But later there is very little discussion on the subject in the results or the conclusions. Furthermore, it is stated that the model is not able to properly represent the decoupling strength (line 297) that is needed for these transitions to occur. So, what can be learned from the results about the importance of decoupling in this environment?

**Specific comments:**

1. Line 40: unclear what "following the polar jet stream" means.
2. Line 45: According to what is written In Sinclair et al. (2010) they find that in the cold sector the PBL is deeper, well mixed and unstable with stronger sensible heat fluxes but is several places (e.g., lines 454-455 and conclusions) it is written that the here you find that the PBL in the cold sector is more stable. This should be clarified, and along these lines, the differences between EIS and stability should be better discussed, as they are not interchangeable. (Also true for the citation of Naud et al (2018a) – line 54).
3. Line 46: define here what negative fluxes mean.
4. Line 47: replace "warm front" by "cold front"
5. Lines 71-78: this paragraph misses more recent works about latest model developments.
6. Line 89 onwards: It is better to start this section by motivating the location and the specific case study. Only then, describe the data in the context of the specific case chosen.
7. Lines 105-123: this is a heavy paragraph that is very hard to read. I suggest separating between variables measured and the instruments in a way that emphasizes the variables that will be later analyzed.

8. Estimation of PBL height: the "eye based"/"best estimate" should be more strictly described how the decision of BL height is made based on the measured or modelled profiles. Why is it not automated based on defined gradient criteria?
9. Lines 130-132: be more specific about the indices and thresholds used, and what do the values indicate? Also, for completeness, note how EIS is defined.
10. Line 148: "semi independent": note which of the above-mentioned observations are assimilated into the IFS model that drives ERA5.
11. Line 169: clarify if shallow convection parameterization is still active for the fine mesh.
12. Line 183 onwards: please motivate the developments elaborated in the next paragraphs.
13. Line 189: missing: q_r is...
14. Line 243: front detection: move the description from section 5.1 to the methods section.
15. Line 244: replace "originate" by "spatially connected"
16. Line 247: are the last 3 days still considered a "post cold frontal region"? and if so, in what sense?
17. Line 256: add "with the eastward propagation of the high-pressure systems" after " southerly component"
18. Line 270-272: how do you account for the horizontal drift of the sondes?
19. Line 286: "underestimates the temperature profile" this is unclear, and not true always/everywhere. It rather smoothes the temperature gradients.
20. Line 288: What does the high vertical resolution simulation show? Is vertical resolution the reason for the poor representation of the inversion?
21. Line 293: "overestimate of entrainment" – where is this seen in the figure?
22. Line 297: Elaborate what do we learn from the indices in Fig. 5? What is a reasonable range of numbers to be considered a good match? it is stated the model does not represent the decoupling well (compared to observations). I'm missing some more explanations on what the implication of this on cloud formation is and the PBL structure in the cold sector in the model.
23. Line 304: what do we learn from the results in this paragraph?
24. Line 330 (figure 8): the description of the moving window is not clear. Are you changing the location of the averaging box in the analysis or keeping it constant in the location of the measurements? Why not look at the changes in time at the same location/gridpoint and make a time series plot corresponding to the observations?
25. In line 323 it is written that "the simulations exhibit a degree of bias relative to the observations, but that bias is only minimally attributed to the differences among the microphysical parametrizations." And in line 349 – "Many but not all of the differences can be explained by the differences in microphysical parametrizations and assumed parameters" – This seems contradicting or does it refer to other differences. Please explain or clarify in the text.

26. Line 338: the results for cloud thickness show a very good match in the histogram (Fig. 8c), why are they so different in Table 2?
27. The method of centering the transect around the front is very interesting and useful. However, if the orientation of the transect is constant, doesn't it imply that some of the expected clouds in the northeastern part closer to the warm front are missed when the transect only "catches" the southern part of the warm sector.
28. Lines 369,374: why not match the field on 850 hPa and the height (currently 1 km) for the front detection? As it is now we expect a mismatch because of the slanting front surface with height. Also, it is unclear in Fig. 9 caption if the front is identified in ERA5 or COAMPS?
29. Line 372: the cyclone center is not visible.
30. Line 412: the location is too far north from the trades, the reference is therefore strange.
31. Lines 430-443: such high latent heat flux values are not uncommon in cold sectors of extratropical cyclones. Need to add relevant references on this.
32. The paragraph starting on line 445 and in the conclusion section attributes PBL height to the surface fluxes. However, the role of dry intrusions from the free troposphere were shown to enhance vertical mixing and PBL deepening by destabilizations, reconciling the strong inversion with enhanced vertical mixing and PBL height in the cold sector.
33. Line 498: at the end of section 5 there is a missing guidance of the reader: what can we learn about the relevance of the mechanisms outlined in the introduction from barotropic environments to the current case?
34. Line 542: Need to discuss the limitations of surface turbulent heat flux parameterizations.
35. Figure 3: what are the green dashed lines? The field of 500-hPa height appear here with no context – need to add this field to Fig. 2.
36. Figure 4: This figure is out of context and can be moved to the supplementary material for smoother reading. Please also add a bar to show the variability among the different model runs. Note errors in the caption of the curve colors, and spell our MAE.
37. Figure 7: which method is used for determining the PBL depth? Use the same tickmarks on both axes in the third panel too. Spell out MAE.
38. Figure 8: the black histogram in panel d is unclear. Are there missing bins?
39. Figure 13: unclear if the fluxes are observed or from ERA5? Which method is used for determining PBL depth?
40. Figure 15: this figure does not add new information that is not evident in the two figures that come before it or it is not clear from the text what it adds. In the text itself regarding this figure I would expect further discussion on the prevailing air streams in the different sectors of the cyclone that actually define the y-axis of the plot.

**Technical comments:**

1. Line 49: the sentence is unclear, please rewrite.
2. Line 276: replace "all" with "and"
3. Line 332: remove minus sign

4. Figure 2: The cold front line is not visible. In caption – remove "t" from "start"
5. Figure 5 caption: remove "n" from "ration"
6. Figure 10: missing panel labels (a,b,…) referred to in the text.
7. Figure 11 caption: what are the vertical lines?
8. Line 466: change "clout" to "cloud"
9. Line 488: fix ".)" to ")."

**References**

Ilotoviz, E., Ghate, V. P., & Raveh-Rubin, S. (2021). The impact of slantwise descending dry intrusions on the marine boundary layer and air-sea interface over the ARM eastern North Atlantic site. Journal of Geophysical Research: Atmospheres, 126(4), e2020JD033879.

Liu, H., Koren, I., Altaratz, O., Heiblum, R. H., Khain, P., Ouyang, X., & Guo, J. (2021). Oscillations in deep-open-cells during winter Mediterranean cyclones. npj Climate and Atmospheric Science, 4(1), 12.

Tornow, F., Ackerman, A. S., Fridlind, A. M., Tselioudis, G., Cairns, B., Painemal, D., & Elsaesser, G. (2023). On the impact of a dry intrusion driving cloud-regime transitions in a midlatitude cold-air outbreak. Journal of the Atmospheric Sciences, 80(12), 2881-2896.

---

## Author Response (AR1)

We thank all three reviewers for their insightful and constructive comments which have made the manuscript much better. Major changes to the manuscript include those that all three reviewers requested that we 1. Clarify our introduction and goals for the paper, 2. Resolve inconsistencies with PBL depth estimation methods, and 3. More clearly present our results and key takeaways. Regarding (1), we have broadened the introduction to include different processes governing cloud behavior in the cold sector of midlatitude cyclones. We have also explicitly stated the gap in knowledge that this paper aims to fill, as well as reframed the goals of the paper. To address (2), we chose a single approach to calculate PBL depth for both the soundings and the model based on an objective, gradient-based method. The calculations of PBL height are now consistent between observations and model throughout the entire manuscript. Regarding (3), We have more harshly interpreted the COAMPS representation of cloud cover and offered key takeaways and suggestions for future model improvement and evaluation. We have also tied our results back to the synoptic environment of the subtropics, something we had promised to do in the introduction but fell short of in the discussion. Furthermore, we had originally intended to leave the frontal identification methods in section 5.1, since we felt this helped the flow of the paper (i.e., having methods and results of the model comparisons in section 4, and then methods and results of the transformed model output in section 5). However, because reviewers #3 and #4 both suggested moving subsection 5.1 to section 2, we acceded to their suggestion and in the revised manuscript have moved subsection 5.1 to a newly added subsection 2.4. Below we address specific reviewer comments point-by-point, with the reviewer comments in red and our responses in black.

**Anonymous Referee #2**

**General comments**
**Clarity and Contextualization:** The introduction lacks clarity in some areas and should define the goals of the paper or contextualize it within existing research more clearly (see SC1 and SC2). The overarching goal of the study is not explicitly stated early on, making it harder to follow the intended contribution. Additionally, the authors should add a stronger connection to previous work, explicitly stating what knowledge gap this study addresses.
We have almost completely redone the introduction as advised by the reviewers, with more emphasis on previous research relating to our study. We also address the knowledge gas this study aims to fill more clearly. Specific improvements can be found in the responses to SC1 and SC2 below.

**Interpretation of Figures & Takeaways:** Some figures contain interpretations that are incomplete or potentially incorrect (SC3). Additionally, discussions of figures often lack clear takeaway points (SC5), making it difficult for the reader to track key findings throughout the manuscript. This also affects the conclusion, as it becomes harder to recall where certain results were discussed.
We have added much more detail on the interpretation of the figures and key takeaways. Specifics can be found in the responses to SC3 and SC5 below.

**PBL Depth Discrepancy:** There is an inconsistency when reporting differences in planetary boundary layer (PBL) depth using two different methods (SC4), and the implications of this discrepancy for commonly used approaches in the field are not discussed.

While there is not a discrepancy with the reported results (the Richardson number from the soundings produces errors of 800 m, while the COAMPS Richardson number produces errors of 400 m), we understand that changing the methods of PBL depth estimation throughout the manuscript is confusing. We have changed the method of PBL depth to be consistent throughout the manuscript and also more accurate than the Richardson number approach. More details on this can be found in the response to SC4 below.

**Framing and Significance of Findings:** One aspect not covered in the specific comments is that the authors may be underselling the importance of their results. For example, the discrepancy between open-cellular convection observed in satellite imagery and closed-cellular convection in the model is potentially significant, given its implications for cloud albedo, climate feedbacks, and solar radiation management. The authors should consider whether these aspects deserve more discussion, either through an expanded interpretation of the findings or by referencing relevant Literature.

We recognize that COAMPS represents the cloud cover poorly, especially in the later period of the simulation, which has motivated us to make some modifications to the analysis. We have removed the last column of Fig. 10, which showed LWP maps from 25 January 2018 12:00 UTC. Around this time, the winds begin to change from northerly to southerly with the approaching high-pressure system, and the observed cloud field transitions from broken clouds to overcast. This transition is not to be confused with the transition to overcast far behind the cold front, but due to the approaching high. To avoid this confusion, we remove all discussions around the northerly to southerly wind (and associated cloud) transition, including why COAMPS struggles to represent the cloud cover in this regime, which will be the topic of a future study. However, the remaining Fig. 10 panels also show poor representation of the satellite cloud field, so we have added more discussion on why we think this is the case (see response to SC5 below).

In addition, in the conclusion we add this discussion:
"Resolved COAMPS cloud fraction is underestimated compared to satellite, but the sub-grid contribution is overestimated. These inaccurate predictions of cloud cover could lead to large model biases in albedo and incoming shortwave radiation (McCoy et al. 2017). Therefore, much work is needed to improve the representation of boundary layer clouds within midlatitude cyclones. We expect that forecasts may improve with improvements to the boundary layer parameterization (Field et al., 2014, Zheng et al., 2024) and horizontal resolution, so as to not overrepresent grid-scale convection associated with instability driven by large surface fluxes. In addition, the balance between large surface fluxes and subsidence needs to be better represented so that the inversion strengths and entrainment into the cloud layer are better predicted."

**Specific comments**
**SC1)** Adding clarity in the Introduction

The second paragraph of the introduction (lines 24–36) is difficult to follow and would benefit from revision to improve clarity. For instance, the sentence *"this tends to decouple the boundary layer, where mixing between the cloud and surface layers is inhibited"* (line 30) is ambiguous. It raises two questions:
1. What exactly does the boundary layer decouple from?
2. Does the sentence imply that mixing is inhibited because the boundary layer is decoupled, or is the decoupling a result of inhibited mixing?
I suggest rewriting this paragraph to provide clearer explanations of the mechanisms involved and how they interact.

The PBL becomes decoupled when the turbulence is not strong enough to mix air throughout the entire depth of the PBL (e.g., decoupling is the result of insufficient mixing). This can occur in a couple of ways: when the PBL deepens and turbulence cannot maintain mixing over the greater depth, or reduced turbulence due to the stabilization of the PBL, as in evaporation of precipitation. We have rewritten this paragraph to better explain the processes that govern boundary layer and cloud properties in the subtropics, as well as the mechanisms that lead to decoupling and ultimately the cloudiness transition:

"In subtropical ocean basins, planetary boundary layer (PBL) and cloud properties are inherently tied to the synoptic environment (Bretherton and Wyant, 1997; Norris and Klein, 2000; Wood and Bretherton, 2006). Conditions in the eastern regions of the semi-permanent high-pressure systems over the subtropical oceans are conducive to the formation of stratocumulus: large-scale subsidence, shallow PBLs, cool sea surface temperatures (SSTs), and weak surface fluxes (Norris and Klein, 2000). Equatorward and westward of the high-pressure center, the stratocumulus clouds transition to shallow cumulus convection. These shallow cumulus have a lower cloud fraction and therefore albedo compared to stratocumulus and are associated with warmer underlying SST, stronger surface fluxes, and deeper, decoupled PBLs (Bretherton and Wyant, 1997). Two main mechanisms are invoked to explain this transition. Bretherton and Wyant (1997) noted the importance of latent heat fluxes deepening and decoupling the PBL. As latent heat fluxes become larger than turbulence driven by radiative cloud-top cooling, the PBL deepens and turbulence no longer sustains mixing throughout the entire depth of the layer, causing the cloud and surface layers to decouple thermodynamically. Conditional instability builds at the surface, allowing shallow cumulus to form while the reduced moisture fluxes leads to stratocumulus breakup and dissipation. Precipitation can also lead to PBL decoupling. Evaporation of drizzle drops from stratocumulus cools and moistens the sub-cloud layer relative to the cloud layer, resulting in a stabilization of the boundary layer (Savic-Jovcic and Stevens, 2008). The stabilization can result in a weakened turbulent mixing and promote a transition to a more cumulus-dominated cloud field."

**SC2)** Make clear the goal of the paper and better contextualize it within current research (Introduction)
The introduction explains the mechanisms relevant to marine low-level clouds and the transitions between different regimes, which are essential points for the reader to understand. However, two critical elements are missing:

**1. The goal of the paper**: It is unclear what the overarching goal of the paper is (is it a proof of concept, are we looking for behaviour that has not yet been analyzed?). For instance, the last paragraph in the introduction alludes to an investigation of multiple "systems" (plural) but as far as I understand it, only a single case study is investigated. The authors should clearly state the aim of the paper upfront to provide readers with a cohesive understanding of the study's purpose and structure. Also consider adding a short sentence regarding these points in the abstract.

The reviewer is correct in that we have alluded to analyzing multiple systems but have only performed a case study on a single system. We have changed the last paragraph in the introduction to make it clear we are only performing a case study. We have also added the overarching goal is to: "develop a new framework for examining the transitions of synoptic, cloud, and boundary layer properties across different regions of the cyclone in both observations and COAMPS, in a way that minimizes errors associated with cyclone strength and phase."

We modify the abstract to: "In this study, we use the Naval Research Laboratory's Coupled Ocean/Atmosphere Mesoscale Prediction System (COAMPS) and an automated cold-front-relative analysis framework to explore the boundary layer structure associated with low clouds across a transect through the cold front of a midlatitude synoptic cyclone."

**2. Contextualization within current research**: The introduction does not adequately explain what is novel about this study or how it addresses a specific gap in knowledge. For instance: What gap in knowledge is being addressed exactly?

We have reconstructed most of the introduction to frame the goal of the paper stated above, focusing on the synoptic elements (subsidence, surface fluxes) that govern the boundary layer structure (depth, degree of decoupling) and cloud cover (stratocumulus to cumulus transition) in both the subtropics and midlatitude cyclones. We add more details on the processes in which regional models struggle. We also more clearly state the novel methodology our study uses.

What is the primary motivation for evaluating the regional numerical weather prediction model in this context?

We have added more motivation for evaluating regional NWP models in the paragraph in lines 71-79.

"The coarse temporal and spatial resolution of the observational network is not able to resolve the detailed structures in a synoptic cyclone (Ghate et al., 2011). While satellite imagery is able to characterize the spatial structure of the cloud field, the vertical structure of the PBL, inversion, and lower troposphere is not well sampled except near upper-air sounding locations. Therefore, regional models, which have much finer spatial and temporal resolution compared to observations, to study PBL clouds in midlatitude cyclones and CAO."

"In summary, models continue to struggle representing boundary-layer cloud-regime transitions, and cloudiness transitions across synoptic systems remain poorly understood, especially in the context of transition hypotheses developed for subtropical cloud systems."

Are midlatitude baroclinic synoptic systems especially important or poorly understood? This last point should be addressed in the paragraph in line 38 to 56.

We have added more motivation for evaluating midlatitude cyclones throughout the introduction.

In the first paragraph: "A large amount of the cloud feedback uncertainty in global circulation models is due to the misrepresentation of marine low clouds over maritime oceans (Bony and Dufresne, 2005; Zelinka et al., 2016, 2020), including that associated with midlatitude cyclones (Bodas-Salcedo et al., 2012)."

Lines 71-79: "However, despite their importance to local weather, marine and aircraft operations, and the global energy budget, both global and regional models struggle to accurately represent the boundary layer clouds and precipitation associated with midlatitude cyclones (Bodas-Salcedo et al., 2012, Field et al., 2014, 2017, Naud et al., 2020)."

**SC3)** Incorrect/incomplete interpretation of figures

I have the following issues with the analysis of results presented in Figures 5 and 7:

1. Line 285: COAMPS only underestimates the temperature profile throughout the lowest 3 km of the atmosphere from 11:30 on 24.01 to 11:30 on 25.01, not for the time frame stated by the authors.

We have updated this sentence in line 285 so that we aren't describing the entire northerly wind condition period but the period suggested by the reviewer.

"For nearly all of the northerly wind period following the passage of the cold front (24 January 11:30 UTC-25 January 11:30 UTC), COAMPS underestimates the temperature profile throughout the lowest 3 km of the atmosphere."

2. The line immediately following: COAMPS only overestimates PBL moisture for the first two subplots of Fig. 5 (24.01 05:30 to 24.01 11:30) and not as stated by the authors.

While it may not be completely obvious by eye, COAMPS does overestimate moisture for a longer time into the cold frontal period (24 January 11:30 UTC-25 January 11:30 UTC) and which we have revised in the manuscript. These small changes in the thermodynamic profiles (both temperature and moisture) demonstrate the large consequences in the cloud base height and cloud liquid water and therefore is important to note.

**(Skipping comment #3 here to answer #3 and #5 together below)**

4. Line 307: not all COAMPS microphysics sensitivity experiments overestimate stability, because unless I have understood something, there are three data points below the 1:1 line in Fig 7 a).

The reviewer is correct in that not *all points* fall above the 1:1 line, but *all schemes*, on average, do underestimate the inversion strength. We have clarified this wording in the manuscript.

"On average, all of the COAMPS microphysics sensitivity experiments underestimate the inversion strength (EIS) but predict it within 1 K (Fig. 7a)."

3. Directly thereafter: to my eyes, the Thompson param. and Kessler param. show similar skill in predicting the decoupling (Fig7c). Please add some sort of metric to support the statement that one is better than the other in this regard.
5. Line 315: "The Kessler parameterization [...] only [overestimates] decoupling for **one** sounding"
→ how are you defining an over/underestimation? Because there is definitely **more** than one Kessler data point above the 1:1 line in Fig. 7c).
We are using the mean absolute error (MAE) to determine which parameterization performs better. We have revised the analysis on the decoupling analysis after changing the method of PBL depth estimation. We have removed the discussion on which parameterization performs better and instead focus on all three parameterizations together, since they all now produce similar results. We also add the significance of COAMPS' representation of the decoupling index.

"All COAMPS simulations, on average, overestimate the decoupling index and struggle to represent the degree of decoupling with errors ranging from 25-100% (Fig. 7c). As described above, this is likely because the model inversion strength is too weak."

**SC4)** Richardson number vs. best estimate approach
In line 130 the authors state that the best estimate approach results in PBL depths that are **800m** larger than those derived from the Richardson number approach. In line 219 the authors put this number at **400m**. Please revise (or in case this is not a mistake, make clear why the difference is 800m and then 400m). Please also discuss the implications this discrepancy has for methods commonly used in the field.
We can see how using two different methods of estimating the PBL depth at different points in the analysis is confusing, which has motivated us to find a different approach to calculate the PBL depth that is consistent throughout the manuscript yet still an apples-to-apples comparison to the observations. Furthermore, the Richardson number threshold approach has been shown to not be a representative estimate of PBL depth and could be one reason for the systematic and well-documented underestimate of PBL depth in COAMPS and other regional models (Wyant et al. 2010, Eleuterio et al. 2004). In well-mixed PBLs capped by a strong inversion, the boundary layer depth can be estimated by the height of the maximum change in temperature, which has been one of the most widely-used methods (e.g., Stull 1988, Liu and Liang 2010). We have decided to use the height of the maximum of the second derivative of the liquid water potential temperature, $(\frac{d^2\theta_l}{dz^2})$. This method matches the previous "best estimate", by-eye approach in the cold sector well (see below figure) and corresponds well with the cloud top height. However, it still struggles, like the Richardson number and many approaches (Seibert et al. 2000), in the more stable frontal and warm sector regions where the PBL is not well-mixed and no capping inversion exists (see below figure). While no singular method accurately represents the PBL depth in all meteorological conditions, approaches using multiple retrievals

have shown improvements (e.g., Roldán-Henao et al., 2024, Smith and Carlin 2024, Zhang et al., 2025).

The newly calculated PBL depths do not change the conclusions, but a few of the results have changed so some of the explanations needed to be revised. The COAMPS PBL depth is now more comparable to the observations. Further, since the COAMPS PBL height increased and is no longer below the cloud layer, the decoupling index has now also increased. This indicates that COAMPS is more decoupled than the soundings. The misrepresentation of the decoupling index is used to explain the misrepresentation of the spatial distribution of cloud in Fig.10.

We added a description of the PBL depth estimation from the soundings in line 126:
"The depth of the planetary boundary layer (PBL) is estimated by the height of the maximum value of the second derivative of the liquid water potential temperature profile ($\partial 2\theta l/\partial z2$), which corresponds to the height of the maximum increases of the $\theta l$ gradient. Although we have not seen this approach used before, it was demonstrated to yield more consistent estimates of PBL depth between observations and the model than methods based on the temperature gradient itself or Richardson number thresholds."

Similarly for COAMPS in line 214:
"The internally computed COAMPS PBL height is determined by a Richardson number threshold of 0.5. The Richardson number approach is robust but tends to underestimate the inversion height by at least 200 m (Wang et al., 2011; Wyant et al., 2010) in cases of a pronounced inversion. Gradient methods have been shown to be more accurate than Richardson number approaches in both soundings and models (e.g., Liu and Liang, 2010; Eleuterio et al., 2004). As described in Sec. 2.1, we estimate PBL depth for both COAMPS and the soundings by the height of the maximum in the second derivative of the liquid water potential temperature."

[Figure]

**SC5)** Emphasize main findings
In the paragraph between lines 398 and 409, as well as the following paragraph discussing Figures 11 and 12a, the authors provide a brief summary of what is shown in the figures. However, the key takeaways are only stated briefly. Clarity would improve if the authors more clearly emphasize the main findings—e.g., by explicitly stating, *'As discussed in Section X, this shows that the model is incapable/capable of [etc.]'* or similar phrasing—to guide the reader and reinforce the significance of the results. For instance, in Line 415–416 (*'Subsidence dominates [the entire cold sector]'*), could you briefly remind the reader whether this behavior aligns with expectations? The first paragraph in page 14 is a positive example and does a good job at this. While most of these takeaways should, of course, be reserved for the conclusion, these modifications would still improve readability. These modifications would also help the reader recall where specific findings were discussed when reaching the conclusion.

We reiterate our results from section 4 that COAMPS produces clouds that are too thick with too much liquid water in the paragraph between lines 398-409. In the same paragraph, we add a discussion on the deficiencies of COAMPS thermodynamic profiles (discussed in section 4, Figs. 5, 7) to help understand the poor LWP representation in Fig. 10.

"The LWP (and associated cloud thickness) as well as the horizontal size of individual cells are much larger than in the satellite, which was also found compared to the ground-based remote sensing observations in section 4 (Fig. 8 and Table 2). This results in a cloud cover that is much less than the satellite. The inversion strength parameter (EIS) has been shown to be well correlated with cloud cover in both the subtropics (Wood and Bretherton 2004) and in the cold sectors (Naud et al. 2016). In this study, COAMPS underestimates the inversion strengths (Figs. 5, 7) which could explain the lack of COAMPS cloud cover. The inability of COAMPS to

decouple the PBL and produce drizzle could explain the overestimation of LWP in the individual cells. Further, the 3 km horizontal resolution may be too coarse to resolve the cellular organization that is shown in the satellite image."

We also add more descriptions of the subsidence properties we would expect behind the cold front when describing Fig. 11 in the paragraph between lines 411-416.
"Subsidence dominates the boundary, inversion, and free tropospheric layers throughout the entire cold sector region, as expected from Naud et al. (2018a) and consistent with the large-scale divergence found in the cold sector by Field and Wood (2007). The deep and turbulent boundary layer capped by a temperature inversion that is sustained by large-scale subsidence is comparable to the environment of the eastern portions of subtropical high-pressure systems where stratocumulus generally form."

**SC6)** Use color deficient friendly color palettes and (perceptually uniform) colormaps
Figures 12b, 13, 14 and 15 use red and green colors on the same plot. This combination is not color-deficient friendly, please a different color palette (see e.g https://www.nceas.ucsb.edu/sites/default/files/2022-06/Colorblind%20Safe%20Color%20Schem es.pdf for a sensible choice of colors) Figure 9 uses a rainbow colormap, which is neither color-deficient friendly nor perceptually uniform. Please change the colormap (viridis, plasma, batlow, etc.).
Thanks for this suggestion. In Fig. 9, we have changed the colormap to 'cividis', a color-deficient friendly colormap.
In Fig. 12b, we changed the color of the latent heat flux transects to dark blue and the color of the sensible heat fluxes to yellow/gold so that the subplot does not contain both red and green. We have changed the frontal sector color from green to yellow/gold in Figs. 13, 14, and 15 so that red and green colors are not in the same plot.

[Figure]

**Figure 9**

[Figure]

**Flgure 14**

[Figure]

**Figure 15**

**Technical corrections**

Please add relevant citations in line(s) 18, 20-22, 25.

We added the citations Paltridge (1974) and Platt (1976) to line 18. Lines 20-22 have been removed after revising the introduction. Line 25 has been modified to: "Much of our understanding of low cloud processes among different cloud regimes has come from observational (Nicholls, 1984; Nicholls and Leighton, 1986; Albrecht et al., 1988; Rauber et al., 2007) and idealized modeling studies (Stevens et al., 2001; Siebesma et al., 2003; Stevens et al., 2005; vanZanten et al., 2011) over barotropic, subtropical regions."

Line 309: make sure you are referencing the correct figure. Also in the rest of the paragraph.
We have double checked, and the paragraph beginning on line 306 describes Fig. 7, as is stated in the manuscript.

Figure 10: the labels "a) b), … i)" are missing
We have added lettered labels to this plot.

Use either "Fig. X" or "Figure X"
We have followed the ACP guidelines of using Figure X at the beginning of sentences and Fig. X elsewhere.

References

Eleuterio, D. P., Q. Wang, and K. Rados, 2004: Diagnostic boundary layer height for cloud-topped boundary layers in a mesoscale model, Research Paper, Naval Postgraduate School, Monterey, CA.

Liu, S. and Liang, X.-Z.: Observed Diurnal Cycle Climatology of Planetary Boundary Layer Height, Journal of Climate, 23, 5790–5809, https://doi.org/10.1175/2010JCLI3552.1, 2010.

Roldán-Henao, N., Su, T., & Li, Z.: Refining planetary boundary layer height retrievals from micropulse-lidar at multiple ARM sites around the world. *Journal of Geophysical Research: Atmospheres*, 129, e2023JD040207. https://doi.org/10.1029/2023JD040207, 2024

Seibert, P.: Review and intercomparison of operational methods for the determination of the mixing height, Atmospheric Environment, 34, 1001–1027, https://doi.org/10.1016/S1352-2310(99)00349-0, 2000.

Smith, E. N. and Carlin, J. T.: A multi-instrument fuzzy logic boundary-layer-top detection algorithm, Atmos. Meas. Tech., 17, 4087–4107, https://doi.org/10.5194/amt-17-4087-2024, 2024.

Stull, R. B.: An Introduction to Boundary Layer Meteorology, Kluwer Academic, 1988.

Wyant, M. C., Wood, R., Bretherton, C. S., Mechoso, C. R., Bacmeister, J., Balmaseda, M. A., Barrett, B., Codron, F., Earnshaw, P., Fast, J., Hannay, C., Kaiser, J. W., Kitagawa, H., Klein, S. A., Köhler, M., Manganello, J., Pan, H.-L., Sun, F., Wang, S., and Wang, Y.: The PreVOCA

experiment: modeling the lower troposphere in the Southeast Pacific, Atmospheric Chemistry and Physics, 10, 4757–4774, https://doi.org/10.5194/acp-10-4757-2010, 2010.

Zhang, D., Comstock, J., Sivaraman, C., Mo, K., Krishnamurthy, R., Tian, J., Su, T., Li, Z., and Roldán-Henao, N.: Best Estimate of the Planetary Boundary Layer Height from Multiple Remote Sensing Measurements, EGUsphere [preprint], https://doi.org/10.5194/egusphere-2024-3959, 2025.

**Anonymous Referee #3**

Major comments:
1. I think the most important part which can be improved is the general motivation on what is new in this particular study and how it fits with earlier studies. There has been quite a lot research already on clouds and precipitation related to midlatitude cyclones. It would be good that the research gap is explicitly stated. In the conclusions the authors mention (lines 566-569) that they "I*n this work, we demonstrate that the analysis method provides a pathway to compare boundary-layer cloudiness forecasts against observations in the context of the different sectors of midlatitude cyclones and in a manner that minimizes errors associated with misplacement of the cyclone center and frontal structures (i.e., phase error).*" If this is the main gap the study tries to address it would be good to explicitly state this in the introduction. Moreover, it would be good to also more explicitly state what 'the analysis method' refers too.

We have almost completely redone the introduction as advised by the reviewers, with more emphasis on the boundary layer and boundary layer cloud properties throughout cyclones, and how PBL processes are represented by regional models, which is the gap in knowledge we are aiming to fill.

In addition, the final paragraph of the introduction has been modified to state the overarching goal of the study, explain why it is novel, and state our two specific goals.

"Understanding the factors driving the evolution of cloud and boundary layer properties in midlatitude cyclones is crucial for improving numerical model forecasts. Here we apply the Naval Research Laboratory Coupled Ocean/Atmosphere Mesoscale Prediction System (COAMPS, Hodur, 1997) regional model to a case study of a wintertime midlatitude cyclone over the Eastern North Atlantic. We develop a new framework for examining the transitions of synoptic, cloud, and boundary layer properties across different regions of the cyclone in both observations and COAMPS, in a way that minimizes errors associated with cyclone strength and phase. We have two linked objectives: 1. to apply our understanding of boundary-layer clouds, largely relevant to barotropic atmospheres in the subtropics, and expand it to boundary-layer clouds accompanying midlatitude baroclinic synoptic systems; and 2. to evaluate to what extent COAMPS represents the synoptic controls on cloud regime and PBL evolution within a midlatitude cyclone, including an analysis on the model's sensitivity to the microphysical parameterization."

In the conclusion, we modify the sentence as the reviewer suggest by adding in what the "analysis method" refers to: "We have composited synoptic, cloud, and boundary-layer properties from COAMPS output and observations relative to the cold front identified by the Naud et al. (2016) automated frontal detection algorithm. We demonstrate that this provides a pathway to compare boundary-layer cloudiness forecasts against observations in the context of the different sectors of midlatitude cyclones and in a manner that minimizes errors associated with misplacement of the cyclone center and frontal structures (i.e., phase error)."

2. Related to this, in the first paragraph two distinct cloud-transition mechanisms are discussed: The deepening warming hypothesis and a cloud transition mechanism that relies on precipitation. Given the first objective of the study (*"to apply our understanding of boundary-layer clouds, largely relevant to barotropic atmospheres in the subtropics, and expand it to boundary-layer clouds accompanying midlatitude baroclinic synoptic systems"*, lines 80-81) some expectations are set that a direct comparison is made between these mechanisms and the case discussed. However, this is never explicitly done. I am not saying the authors should definitely do this, however I still find it hard to understand what the conclusion is regarding this first objective after reading the manuscript. This is also because in the conclusions the main focus is on the results regarding the second objective. It would be good to more explicitly state the conclusions regarding this objective.
We have modified the second paragraph in the introduction to discuss the synoptic and boundary layer properties, specifically decoupling, governing cloud cover in the subtropics, rather than focusing on the cloud transition mechanisms (see response to reviewer 2 SC1). While this paper does point out the transition from open cellular clouds just behind the cold front to stratocumulus further behind the cold front and closer to the high-pressure system, we do not explicitly address the mechanisms of this transition. We have added a paragraph in the conclusions to discuss the results of the first objective, specifically how our results tie in to paragraph #2 of the introduction.

"This study has revealed the synoptic, boundary layer, and cloud properties along a transect from far (~1500 km) behind the cold front, across the cold front, and into the warm sector. In the cold sector, a continuum of cloud and boundary layer properties exist, as in the subtropics. The environment far behind the cold front is most reminiscent of the region of the eastern subtropical ocean basins and is characterized by strong subsidence (~3 cm s-1) and stability across the inversion (EIS of ~11K). Moderate values of surface heat fluxes (150 W m−2 latent and 50 W m−2 sensible) lead to moderate PBL depths (~1.3 km). The lower PBL depths in this region allow for the boundary layer to be more readily coupled (i.e., nearly well-mixed) than any of the other regions. This region, like the subtropics, has nearly homogeneous cloud cover. In our study with 3 km horizontal grid spacing, most of that cloudiness is on the sub-grid scale. Within 750 km of the cold front, the surface fluxes are a maximum (300 W m−2 latent and 100 W m−2 sensible). The PBL is as deep as~1.5 km and becomes slightly more decoupled. If present, additional dry air entrained into the top of the PBL via the dry intrusion mechanism could be aiding in increasing the turbulence to deepen and decouple the PBL in this region with

decreased subsidence and weaker inversions, as suggested by Ilotoviz et al. (2021) and Tornow et al. (2023). Closer to the front, in an environment reminiscent of where the stratocumulus transition to cumulus in the subtropics, the cloud cover decreases to about 0.6 and the LWP increases to 100 g m−2. The frontal sector is characterized by upward vertical motion, resulting in forced convective clouds with high LWP (> 100 g m−2). The inversion strengths (5 K) and surface fluxes (100 W m−2 latent and 0 W m−2 sensible) are much weaker than the cold sector. The warm sector has the weakest surface fluxes (50 W m−2 latent and 0 W m−2 sensible), owing to the shallowest (< 500 m) and most stable PBLs with the lowest cloud cover (< 0.2) and LWPs (< 20 g m−2). Weak vertical motion results in weak inversion strengths (< 1 K)."

3. The section on how the cyclones are detected and how the cross-frontal composites are made (section 5.1) is quite detailed, I would suggest to move that to the methods section, since that is also the section where the reader expects this information. Moreover, it is already used before in e.g. Figure 2.

At the reviewer's suggestion, we have moved the discussion on cyclone center and frontal identification algorithms (lines 365-372, 376-382) from section 5.1 to a newly created subsection 2.4 (2.4 Cyclone center and front identification). However, we have kept the discussion of Fig. 9 and the description of the transect and sector definitions (lines 372-386, 382-396) in section 5. Section 5 no longer has any subsections.

4. The presentation of the figures is already much better with the corrections made, however could still be improved sometimes, see detailed comments below.

We have made several changes to the figures and figure captions to improve clarity as suggested by the reviewer. Specific improvements are noted in the responses to comments #9-17 below.

Minor comments:
1. Abstract line 1: It might be my ignorance, but can you really put this so strongly? Or do you refer mainly to the knowledge in a theoretical (boundary layer) framework? I think e.g. there is e.g. quite a lot of studies already e.g. on mesoscale rainbands already decades ago (e.g.Houze et al., 1976, Matejka et al. 1981, Knight and Hobbs, 1988), and there are also quite some empirical studies on clouds associated with fronts and extratropical cyclones as well.

While there has been research on frontal cloudiness associated with the cold front and warm fronts as well as the various conveyor belts associated with the cyclone, none of the papers cited by the reviewer emphasize boundary layer clouds. This study focuses on how the boundary layer structure controls the formation and evolution of boundary layer clouds, specifically in the cold sector, which has been relatively understudied.

2. Line 8: "The Frontal region" does this refer to the cold or warm front or the fronts in general?

This refers to the cold frontal region. We have added "cold" to this sentence in the abstract to clarify.

3. Line 41: Given the broad scope of the study and possible broad background of the readers, it might be good to introduce the regions of there one would large-scale vertical ascent in a typical extratropical cyclone.

We have added more detail into the expected synoptic properties, including where large-scale ascent and descent typically occur in extratropical cyclones. This is added in the paragraph in lines 38-56.

4. Line 90-95: I think the discussion on the summer cloud properties in the Azores region is not strictly necessary and might be even be confusing, since this study focuses on a week in winter.

We agree. We have removed the discussion on summer synoptic and cloud properties.

5. Line 100-104: I do not completely see why a positive NAO phase would not be representative for winter storm systems. I would earlier maybe state the opposite, since the negative phase is more associated with a blocked weather pattern over Europe.
Furthermore, I would also argue that the position of the studied extratropical cyclone matches quite well with the average position of the storm tracks (see e.g. Neu et al., 2013). Therefore I would suggest to rewrite this part a bit.

After double checking, the reviewer is correct that the NAO does not seem to have affected the position of the cyclone in this study, and that it is located within the area of highest density of storm tracks in several studies. We have modified this part in the manuscript:
"We note that January 2018 was in the positive phase of the North Atlantic Oscillation, so both the Icelandic Low and Bermuda High were stronger than normal, which tends to shift the storm tracks northward (Hurrell, 1995). However, the cyclone center in our case seems to follow the average wintertime North Atlantic storm tracks in Neu et al. (2013) and Wang et al. (2023)."

6. Line 152: It might be good to already introduce the exact period of the model simulations, or remove this part completely here, since now it is scattered over two paragraphs.
We reintroduce the study period in line 152.

"Simulations of the study period (22-29 January 2018) are performed using NRL COAMPS (Hodur 1997)."

7. Line 217: "Only over the ENA site" Which region is chosen for this?
We had previously only calculated the best estimate of PBL height by hand over the average thermodynamics in the analysis box in Fig. 1, in order to calculate the minimum number by hand. However, we recognize that having two different methods of PBL depth estimation throughout the manuscript is confusing and have since changed the methodology to be consistent throughout (see response to reviewer 2's SC4).

8. Line 217-219: Given the use of the same Richardson constant of 0.5, do the authors have an idea why the underestimation in the regional model is only by about 400 meters, compared to the 800 meters using the sounding data?

The COAMPS Richardson number depth is usually near the cloud base height, above which is a slightly stable cloud layer where the Richardson number drastically increases with height. This results in an underestimate of the inversion height by approximately the cloud thickness (~400 m). However, the soundings are much less smooth than COAMPS due to their enhanced resolution. At night, they also have a weak temperature inversion at the surface (see Fig. 4 potential temperature profiles) that quickly increases the Richardson number profile, as the Richardson number separates this stable layer from the well-mixed layer above. For these reasons, the height at which the Richardson number becomes more than 0.5 is often much lower in the atmosphere than COAMPS.

In both the soundings and COAMPS, the Richardson number seems to be doing a poor job estimating the height of the inversion, as found by other studies (e.g., Eleuterio et al. 2004). For this reason, we have chosen a different approach that greatly improves our estimation of PBL depth.

9.  Line 243-245: The system is already occluded probably, though it is hard to see from the chosen region, which might be the reason why your front is not originating at the centre. The phrasing suggests that this is not visible in the temperature fields, if this is not the case I would rewrite this part. Furthermore, as far I understand you are using a frontal detection at 1 km? Given the (potential) tilt of fronts with height, one would not always have 100 % correspondence anyways.

Yes, we agree that the system appears to be occluded, which is why the front does not necessarily originate at the main low pressure center. It does, however, originate from a secondary low formed at the triple point of the cold, warm, and occluded fronts. This can now more easily be seen by extending the plots northward. In addition, we note that the frontal position *is* consistent with where the front should be placed given the near-surface (1 km) temperature field (line 244).

We modify line 243 onwards: "The cyclone appears to be occluded throughout the period, with the cold fronts originating from the secondary low formed from the frontal instability at the triple point (Schemm and Sprenger 2015)."

10. Lines 382-385/Figure 9: Why is the transect not centred in the middle of the cold front? And how is the exact location along the cold front determined?

The transect is stationary for the entire study. The location of the transect was chosen so that it intersects the ENA observational site, to allow for easy comparisons between COAMPS and the observations. The orientation is perpendicular to the cold front. As the cold front moves SE, the point it intersects with the transect is manually identified as distance 0 in Figs. 11, 12.

11. Figure 2: Can the displayed region be extended a bit to the north? I understand that the main region of interest are the frontal regions, but now the studied cyclone is not (completely) visible at displayed time steps.

We have increased the northward extent of the plots to 65N, which better shows the entire low center. It also better shows the occlusion of the cold front, as suggested by comment #9, which we have addressed in the manuscript (see response to comment #9).

12. Caption Figure 2: "Red start" should be "Red star"
Thanks for catching this. We have modified the caption to "star" instead of "start".

13. Figure 4: Over which region is the MAE calculated?
The MAE is calculated over the entire coarse mesh, regridded to the ERA5 grid, for each forecast time. This is stated in lines 276-277.

14. Figure 5: I also would suggest moving the legend out of the plotting area here. Moreover, the legend is also not completely clear to me, does it refer to the lines plotted or the coloured squares indicating the cloud boundaries?
The legend describes both the lines and squares. Both the green line and box are COAMPS, the sounding is the black line, and the gray box is the ARSCL cloud boundaries. We have modified the legend to include all lines and boxes and moved it outside of the plotting area.

15. Figure 8: The densities of panel (e) are beyond the y-axis. Futhermore I am a bit confused on that the total density (surface area below the lines) does not seem to add up to the same value for each subplot. Is this due to the different number of observations points in each of the subpanels?
The number of samples is different between each scheme (i.e., Kessler has a different number of samples than KK Lite), so each subplot is normalized. The LWP, CTH, rain rates, and evaporation rates are all cloud-conditioned, so each individual scheme has the same number of samples in these subplots. However, cloud cover is not cloud-conditioned and therefore has a different number of samples than the other subplots. We have confirmed that the area under each curve is 1 (see figure of CDFs). We also extended the y-axis limits of panel (e), and we hope that this makes it easier to see that these curves do all have an integral of 1.

Aside from panels (c) and (e), we think that the reviewer may be distracted by the logarithmic axis, which does make the tails appear to contribute more to the total density than they actually do. To show this, we have plotted the CDFs of each to show the contributions from each bin to the total, confirming they do all add up to the same number and that the larger values of LWP and precipitation rates contribute relatively little to the total.

[Figure]

16. Figure 10: I understand the choice of a blue colour to represent of the cold front, however given the blue colour map in most panels this makes it hard to distinguish from the background. I would suggest making the location of the front black.

We have changed the color of the front position to black in Fig. 10.

17. Figure 11, caption: I wondered here why the authors to use the median here, since using only 8 forecasts might result in a skewed result. Have the authors compered to e.g. using the mean to see if the results would be very different? Moreover, is the median calculated at each point separately?

The medians are calculated between the 8 transects in 100 km bins (line 411). We modified the manuscript so that this is also stated when describing Fig. 11 and in the Fig. 11 caption.

The average (mean) of the 8 transects is not too different from the median (average of the transects shown below). The average and median water vapor and potential temperature transects are nearly exactly the same. There are only slight differences in the vertical motion

field, with a stronger, narrower region of ascent right at the cold front, and stronger accent between ~200-400 km behind the cold front above the boundary layer.

Looking at the Fig 12b,c individual transect lines, it's not too difficult to see that the mean and the median of all 8 transects within a given bin is the same, as the change between each of the transects is nearly linear. The range in values is due to the differing stages of the cyclone life cycle at the different transect times.

We have added to the Fig. 11 caption: "We choose to analyze the median of the 8 transects but using the mean does not change the results."

[Figure]

**Anonymous Referee #4**

Major comments:
1. Aims of the current study (lines 80-86): the aims as written in this paragraph do not reflect what is later achieved. Aim #1 does not apply knowledge on midlatitude systems, but only to one case study. Aim #2 is a methodological plan, but it is unclear what is the more specific scientific objective it aims to address? (e.g., examine parameterizations, assess operational NWP skill? This is unclear).

Multiple reviewers had criticisms of our motivations and goals, which we have endeavored to address in the revised manuscript. We have modified our goals so that it is clear we are performing a case study and not analyzing multiple systems. Aim #2 has also been modified and now reads, "to evaluate to what extent COAMPS represents the synoptic controls on cloud regime and PBL evolution within a midlatitude cyclone, including an analysis on the model's sensitivity to the microphysics parameterization."

2. Missing consideration of cyclone airstreams in the baroclinic environment: The introduction and discussion should acknowledge the importance of cyclone-related airstreams for shaping the cyclone environment, including the PBL regimes and clouds. Recent papers address these issues. Specifically for ENA, Ilotoviz et al. (2021) highlighted the role of dry intrusions for shaping cold-sector PBL regimes and clouds. Turnow et al. (2023) note the importance of rain onset for cloud transitions using LES, explaining differences among cold sector regions closer Vs. away from the cyclone center. Liu et al (2021) studied cloud transitions in post-cold fronts regions of Mediterranean cyclones, relating the transitions to precipitation dynamics. Vast literature relates warm conveyor belts to clouds along the cold front and in the warm sector, while other studies relate surface fluxes to the passage of cyclones and/or cold fronts. Since one of the aims of the current study is to examine whether known PBL regimes apply to baroclinic environments, these aspects should be better introduced and discussed again in this context.

We have revised the discussion on cyclone properties (lines 38-70), centering it around the impacts of the airstreams found throughout the cyclone, including the warm conveyor belt, cold conveyor belt, and dry intrusions. We cite Ilotoviz et al. (2021) and Tornow et al. (2023) results to explain the cloud and boundary layer properties in the cold sector. We also add sentences to the conclusion on the role dry intrusions may play in our study.

"Tornow et al. (2023) found that earlier onset of rain formation triggered earlier transitions from overcast to broken clouds during cold air outbreaks (CAO) in the cold sectors of midlatitude cyclones, similar to the precipitation-driven transitions in the subtropics (e.g., Savic-Jovcic and Stevens, 2008). In some situations, slantwise dry-air intrusions in the cold sector have been shown to additionally influence cloud and boundary layer properties through increases in surface fluxes and associated turbulence, which can deepen the PBL and increase the cloud cover on average (Ilotoviz et al., 2021; Tornow et al., 2023)."

"If present, additional dry air entrained into the top of the PBL via the dry intrusion mechanism could be aiding in increasing the turbulence to deepen and decouple the PBL in this region with

decreased subsidence and weaker inversions, as suggested by Ilotoviz et al. (2021) and Tornow et al. (2023). Though our results are consistent with the presence of a dry intrusion, without an extensive back trajectory analysis, we cannot definitively say that the cyclone studied is influenced by dry intrusions and could just be regular post frontal conditions."

3. Overall, the representation of boundary-layer clouds in COAMPS is not good (Fig. 10 mainly, but also in terms of cloud cover etc.). Therefore, the notion in lines 506-507 should be toned down. I was left wondering what can we still learn from the results of the evaluation study. For example, one of the key finding is the relatively small influence of the microphysics scheme on cloud parameters (which somewhat raises the question why is this aspect so prominent in the methods section?). Given the model deficiencies, what do we learn more specifically that can guide future model developments?

While the reviewer is correct that Fig. 10 does not reproduce the satellite image well, it is important to find different ways of evaluating the cloud properties besides instantaneous comparisons of spatial configuration (e.g., Mass et al. 2002), which is what we have done throughout most of the analysis. The distributions of synoptic and cloud properties across the different sectors as a whole (Figs. 13, 14) do reveal good comparisons to observations. Further, we have removed the right column of Fig. 10 (c,f,i) which shows the LWP from 20180125 12 UTC. At this time, the high-pressure system is dominant and the winds are southerly, which is not the period we are focused on. We are also not interested, in this paper at least, in this transition to stratocumulus, and therefore we will not discuss why COAMPS struggles with this regime.

We add "...*somewhat* credibly represents…" to tone down the notion in lines 506-507.

However, there is definitely room for improvement, and we could evaluate every single parameterization, but that would be a lot of work for a single study. We chose to focus on one aspect of the model - the microphysics parameterization. We used the best microphysics scheme available to us and although it did not provide as much benefit as we had hoped, the results are still insightful especially in that they suggest what should be a microphysical parameterization better suited for boundary-layer clouds may not automatically result in a vastly improved prediction. In addition, we already have suspected that the clouds are more sensitive to the boundary layer parameterization (line 549), but we expand on this conclusion.

"The clouds likely are more sensitive to the boundary layer parameterization (Juliano et al. 2019). Lamraoui et al. (2019) found that the degree of decoupling is sensitive to the PBL scheme, but modeled cloud fraction is more sensitive to the shallow convection scheme. However, evaluating the PBL and shallow convective scheme sensitivity in COAMPS is left for a future study."

4. In the introduction (Lines 24-36) there is a long discussion on the transition mechanisms between the different cloud fields regimes. But later there is very little discussion on the subject in the results or the conclusions. Furthermore, it is stated that the model is not able to properly

represent the decoupling strength (line 297) that is needed for these transitions to occur. So, what can be learned from the results about the importance of decoupling in this environment?
We have modified the discussion in lines 519-542 in the conclusion to address the cloud and boundary layer regimes and tie our results back to the barotropic environment described in the introduction. We focus on the importance of decoupling and atmospheric stability in the discussion.

Specific comments:
1.   Line 40: unclear what "following the polar jet stream" means.
Regions of jet streaks have enhanced baroclinic instability and thus play a large role in governing the path and evolution of cyclones (Uccellini and Johnson 1979).
In revising the introduction, this has been removed.

2.   Line 45: According to what is written In Sinclair et al. (2010) they find that in the cold sector the PBL is deeper, well mixed and unstable with stronger sensible heat fluxes but is several places (e.g., lines 454-455 and conclusions) it is written that the here you find that the PBL in the cold sector is more stable. This should be clarified, and along these lines, the differences between EIS and stability should be better discussed, as they are not interchangeable. (Also true for the citation of Naud et al (2018a) – line 54).
The reviewer is correct in that we have not been consistent with our definition of stability throughout the manuscript. The instability in Sinclair et al. (2010) and Naud et al. (2018) refers to the internal stratification of the PBL, where instability is associated with enhanced turbulence in the PBL. This is true for our case as well, but we assess the stability of the PBL using the decoupling index, where a smaller decoupling index indicates mixing strong enough to maintain a well-mixed PBL, and a larger decoupling index indicates inhibited mixing between cloud and sub-cloud layers. We use EIS to determine the stability across and strength of the inversion. Here, larger EIS values indicate a stronger temperature jump across the inversion and reducing mixing across the inversion. We have added a discussion on the differences between EIS and PBL stability in line 130. We have also changed the language throughout the manuscript when we are discussing EIS. Instead of "stability (EIS)", we use inversion strength (EIS) or "stability across the inversion (EIS)".

"EIS measures the strength of the thermodynamic jump across the inversion and has been shown to have a positive correlation to cloud cover, when averaged over appropriate timescales, in both the subtropics (Wood and Bretherton, 2006; De Szoeke et al., 2016) and cold sectors of midlatitude cyclones (Naud et al., 2016)."

3.   Line 46: define here what negative fluxes mean.
We have defined negative heat fluxes as "energy transported from the atmosphere to the ocean".

4.   Line 47: replace "warm front" by "cold front"
We have corrected this typo.

5. Lines 71-78: this paragraph misses more recent works about latest model developments.
We have added much more detail describing recent works about PBL clouds associated with cold air outbreaks and midlatitude cyclones in models (e.g., Zheng et al. 2024).

"More sophisticated treatments of model physics have been yielding improvements to PBL forecasts. Zheng et al. (2024) was able to qualitatively simulate the overcast to broken cumulus transition using 3 km grid spacing in a global cloud resolving model, including the increases in surface fluxes and deepening of the PBL which leads to stratocumulus breakup and deeper shallow convection. However, the liquid water was still underestimated, and the mesoscale organization was not well-represented compared to satellite. In summary, models continue to struggle representing boundary-layer cloud-regime transitions, and cloudiness transitions across synoptic systems remain poorly understood, especially in the context of transition hypotheses developed for subtropical cloud systems."

6. Line 89 onwards: It is better to start this section by motivating the location and the specific case study. Only then, describe the data in the context of the specific case chosen.
It seems that this is already what has been done. The first paragraph of section 2 motivates the location and timing of the case study. The second through fifth paragraphs describe the observational data used. We are unsure how specifically to address this comment, but we hope our extensive reworking of the introduction and transition into the data and methods sections addresses these concerns.

7. Lines 105-123: this is a heavy paragraph that is very hard to read. I suggest separating between variables measured and the instruments in a way that emphasizes the variables that will be later analyzed.
We modify this paragraph so that the instrument properties and data products are all listed first, and then focus on the variables that are used for the analysis.

8. Estimation of PBL height: the "eye based"/"best estimate" should be more strictly described how the decision of BL height is made based on the measured or modelled profiles. Why is it not automated based on defined gradient criteria?
We have changed the method of PBL depth, based on a gradient approach, so that the depth is now automated and determined by objective criteria (see response to Reviewer 2 SC4).

9. Lines 130-132: be more specific about the indices and thresholds used, and what do the values indicate? Also, for completeness, note how EIS is defined.
We have added the equation for EIS from Wood and Bretherton (2006) in line 130. As stated in comment #2, we have added a discussion on the interpretation of the various parameters (decoupling index, EIS) and thresholds.

Line 130 to the end of the paragraph now reads:
"We calculate the decoupling indices using the two methods described in Jones et al. (2011) based on the differences between the upper and lower PBL liquid water potential temperature

$(D_\theta)$ and total water mixing ratio $(D_q)$ to determine the amount of turbulence in the boundary layer. A boundary layer that is not decoupled $(D_{\theta,q} < 0.5\ K,\ g\ kg^{-1})$ is well-mixed with a neutral thermodynamic profile, and a decoupled boundary layer $(D_{\theta,q} > 0.5\ K,\ g\ kg^{-1})$ has cloud properties that are thermodynamically distinct from the surface because turbulence is not strong enough to mix through the entire layer. When we use
the term coupled or decoupled, we are referring to these specific thresholds defined in Jones et al. (2011), acknowledging that in reality coupling and decoupling are a matter of degree and not a binary state. Finally, we calculate the estimated inversion strength (EIS) from Wood and Bretherton (2006). EIS is defined by the equation: $EIS\ =\ LTS\ -\ \Gamma_m^{850}(Z_{700}\ -\ LCL)$, where LTS is the lower tropospheric stability (Klein and Hartmann 1993), which is defined as $\theta_{700}\ -\ \theta_{1000}$. We use the potential temperature at 1000 hPa instead of the surface to avoid the strongly stable surface layers that are present throughout most of this study. $\Gamma_m^{850}$ is the moist adiabatic lapse rate at 850 hpa, calculated from the temperature and moisture at 850 hPa (equation 5, Wood and Bretherton 2006). $Z_{700}$ is the height of the 700 hPa surface, and finally, the LCL is the height of the lifting condensation level. EIS measures the strength of the thermodynamic jump across the inversion and has been shown to have a positive correlation to cloud cover, when averaged over appropriate timescales, in both the subtropics (Wood and Bretherton, 2006; De Szoeke et al., 2016) and cold sectors of midlatitude cyclones (Naud et al., 2016)."

10. Line 148: "semi independent": note which of the above-mentioned observations are assimilated into the IFS model that drives ERA5.
We have decided to remove "semi-independent" from the paper as this seems to have added confusion. What we were referring to was the potential for GFS and IFS to be constrained by many of the same datasets. In any case, without extensive detective work we have no comprehensive way of knowing what data is assimilated into either model.

11. Line 169: clarify if shallow convection parameterization is still active for the fine mesh.
We have added that the shallow convection parameterization is active in the fine mesh.

"The shallow convection parameterization is active in both meshes."

12. Line 183 onwards: please motivate the developments elaborated in the next paragraphs.
We have added the following motivation to the beginning of the paragraph in line 183:

"The Kessler (1969) scheme is simple with several arbitrary parameters (Morrison et al., 2020), whereas the Khairoutdinov and Kogan (2000) (KK2000 hereafter) has parameters derived from nonlinear regression of bin-microphysics spectra taken from large-eddy simulations of shallow clouds. For this reason, we supplement the Kessler (1969)..."

13. Line 189: missing: q_r is…

We have moved the definition for q_r from line 194 to line 189.

14.  Line 243: front detection: move the description from section 5.1 to the methods section.
We have moved the description of the front detection methods to the methods section (new section 2.4; see also response to Reviewer 3, major comment 2).

15. Line 244: replace "originate" by "spatially connected"
We have made the wording change as suggested by the reviewer.

16. Line 247: are the last 3 days still considered a "post cold frontal region"? and if so, in what sense?
The last 3 days are not considered post cold front. In this study we refer to the cold sector as only regions with northerly winds. This northerly wind period ends at ENA at 25 January 2018 18:00 UTC, when the winds begin to have a southerly component (Fig. 3d, line 263-264). In the model, the fine mesh begins to contain regions with southerly winds after HR 54, which is why we don't include transects after HR 54 in the analysis (line 394).

17. Line 256: add "with the eastward propagation of the high-pressure systems" after " southerly component"
We have added to this sentence as the reviewer suggested.

18. Line 270-272: how do you account for the horizontal drift of the sondes?
Throughout the study period, the maximum horizontal drift in the lower 3 km is 7 km, with an average of about 4 km. This is much smaller than the scale of the synoptic features we are interested in. Fronts generally have a horizontal:vertical slope ratio of about 50:1 km, which is also much greater than the drift, so it is unlikely that the radiosondes could be sampling different synoptic environments due to their drift, and therefore the drift is not taken into consideration.

19. Line 286: "underestimates the temperature profile" this is unclear, and not true always/everywhere. It rather smoothes the temperature gradients.
We have clarified and added detail into exactly where the model under/overestimates the temperature and moisture profiles (see also response to Reviewer 2 SC 3.1, 3.2).

20. Line 288: What does the high vertical resolution simulation show? Is vertical resolution the reason for the poor representation of the inversion?
Yes, insufficient resolution is likely a reason for the lack of a sharp inversion (as could be an underestimate of subsidence). However, as we have discussed in lines 314-315, decreasing the vertical grid spacing to 25 m did not noticeably yield a better representation of the inversion, suggesting that 25 m is probably still coarse. Unfortunately, substantially improving the vertical resolution takes us out of the realm of workable operational numerical weather prediction and therefore out of the scope of this study.

21. Line 293: "overestimate of entrainment" – where is this seen in the figure?

We determine entrainment is too strong by inference. An overestimate of the entrainment of warm, dry free tropospheric air into the boundary layer results in an underestimate of boundary layer moisture and overestimate of temperature. Entrainment acts to raise the height of the inversion, so an overestimate of the height of the inversion is consistent with an overestimate of entrainment (Bretherton and Wyant 1997). Weak inversions lead to decreased stability across the inversion, which supports increased mixing (Galewsky 2018).

We add these details to line 293:
"The underestimate of moisture, overestimate of inversion height, and weak inversion strengths are consistent with an overestimate of entrainment in the model perhaps deriving from the crude vertical resolution at the inversion."

22. Line 297: Elaborate what do we learn from the indices in Fig. 5? What is a reasonable range of numbers to be considered a good match? it is stated the model does not represent the decoupling well (compared to observations). I'm missing some more explanations on what the implication of this on cloud formation is and the PBL structure in the cold sector in the model.

In section 2.1, we add a discussion on the thresholds used for the decoupling index and what they mean (see response to comment #9). We have also added a discussion on how the prediction of the decoupling index impacts the cloud base height and other profile properties.

"Although Remillard et al. (2012) showed that ENA is in a near constant state of decoupling, the radiosondes only indicate a decoupled PBL (D,q > 0.5 K, g kg-1) for the first 24-30 h after the cold front passage. While the COAMPS decoupling index values are in good agreement with the sounding indices during this period, COAMPS does not actually stratify the temperature and moisture profile as in the observations (Fig. 5). The reason for the larger COAMPS indices is because the top of the PBL lies within the inversion and therefore the upper PBL is warmer than the surface. The discrepancy between the ARSCL and COAMPS cloud base heights may also be explained by the discrepancy in the decoupling. In coupled boundary layers, the lifting condensation level (LCL) and cloud base are at nearly the same height, but in decoupled boundary layers, they may diverge by several hundred meters due to drier air in the cloud layer (Jones et al. 2011). In the soundings, entrainment appears to warm and dry the cloud layer compared to the surface, resulting in a decoupled cloud layer and an LCL that is ~300 m lower than the cloud base (not shown). In COAMPS, the entrained free tropospheric air is much cooler than the soundings and therefore does not decouple the cloud layer but keeps it well-mixed, resulting in a cloud base that is near the LCL. After 30 h, the height of the inversion steadily decreases, and both COAMPS and the soundings maintain a well-mixed profile for the remainder of the northerly wind period, suggesting that the PBL thermodynamic structure in cyclones is highly transient compared to ENA conditions during more quiescent, periods (Remillard et al. 2012)."

While this does also have impacts on cloud cover, addressing that at this point in the paper would be premature, as maps and analysis of the cloud field has not been shown yet. Instead,

we refer back to this point in the beginning of section 5 when discussing the lack of clouds in Fig. 10 (see response to reviewer 2 SC5).

23. Line 304: what do we learn from the results in this paragraph?

In order to explain boundary layer evolution, we need to take into account all factors that contribute to it, which includes the thermodynamic properties (Fig. 5) as well as the dynamic properties (Fig. 6). It is stated that the COAMPS winds are well predicted, and thus the evolution of a decreasing PBL depth with decreasing PBL wind speeds throughout the period is consistent between the radiosondes and COAMPS.

We add: "Discrepancies in the representation of the PBL structure between COAMPS and the soundings (Fig. 5) do not appear to be attributable to the PBL winds."

24. Line 330 (figure 8): the description of the moving window is not clear. Are you changing the location of the averaging box in the analysis or keeping it constant in the location of the measurements? Why not look at the changes in time at the same location/gridpoint and make a time series plot corresponding to the observations?

The spatial sampling window in COAMPS does not move. It is a 144x144 km2 box centered at Graciosa Island, which is the location of the observations. The COAMPS analysis box "sees" farther than the observations (because the observations are taken at a single point), so to make up for this, the observations are sampled for a longer period of time to account for advection through the analysis box.

To attempt to make this clearer, we have added to line 331:
"The observational sampling window is 4 h (+/- 2 h) longer than COAMPS to account for advection (assuming a 10 m/s wind speed), through the COAMPS analysis box."

We could create a time series, but we are less concerned with how COAMPS represents the properties of the observations 1:1 directly at an exact time, and more so concerned with the distribution of properties over a longer period of time. This is a more realistic comparison to models, as we don't (and shouldn't) expect models to exactly align with observations (e.g., Mass et al. 2002).

25. In line 323 it is written that "the simulations exhibit a degree of bias relative to the observations, but that bias is only minimally attributed to the differences among the microphysical parametrizations." And in line 349 – "Many but not all of the differences can be explained by the differences in microphysical parametrizations and assumed parameters" – This seems contradicting or does it refer to other differences. Please explain or clarify in the text.

In line 328, we use the term "bias" to describe the difference between the observations and COAMPS. In an attempt to minimize this bias, we perform many sensitivity simulations with different microphysics schemes with different parameters (Table 1). Though these schemes all produce different results, none of them help to minimize biases in all variables (i.e., the bias between model and observations is greater than the differences among the different model

sensitivity experiments). We want to understand the differences among the simulations to make sure that all schemes produce relationships between variables that are physically consistent and in line with expectations, which is what we discuss in lines 348-362.

We clarify in line 349:
"Many but not all of the differences between simulations can be explained by the differences…".

26. Line 338: the results for cloud thickness show a very good match in the histogram (Fig. 8c), why are they so different in Table 2?

The observations have a larger density of small thickness values (0-200m). The Thompson and KK Lite have nearly an order of magnitude fewer points with 0-200 m thicknesses, but have higher densities of larger thickness (500-1500 m) clouds, which results in a larger mean. This is more easily seen when we plot the observations on top of the COAMPS schemes, instead of the black observation line being obscured by the colored COAMPS lines (see below updated version of Fig. 8).

[Figure]

27. The method of centering the transect around the front is very interesting and useful. However, if the orientation of the transect is constant, doesn't it imply that some of the expected clouds in the northeastern part closer to the warm front are missed when the transect only "catches" the southern part of the warm sector.

Yes, the transect "misses" the cloud close to the warm front, but we were never going to capture those anyway because they are outside the model domain. The warm sector region that we are catching is indeed very far south of the warm front but is still associated with the warm sector.

We have added clarification to what is meant by the warm sector properties in line 396.

"In this case, the warm sector region is far south of the warm front, so the transect captures the properties of the warm sector but not those associated with the warm front itself."

28. Lines 369,374: why not match the field on 850 hPa and the height (currently 1 km) for the front detection? As it is now we expect a mismatch because of the slanting front surface with height. Also, it is unclear in Fig. 9 caption if the front is identified in ERA5 or COAMPS?

According to Naud et al. (2016), the two frontal detection methods (Hewson (1998) based on 1-km potential temperature and Simmonds et al. (2012) based on 850-mb wind) are used in conjunction to find a best estimate of the frontal position. In the case that both methods identify the front in different positions, more tests and processing are applied to determine the best position (see Naud et al. 2016 Fig. 1c).

The 850-mb geopotential height is about 1.5 km, so while changes to the frontal position between each method could arise due to the slant with height, these changes are expected to be minimal (~25 km if assuming a 50:1 km slope) and not substantially change our results.

The frontal position is identified from COAMPS data. This was explicitly stated in line 372, but we also modified the caption so that this is clear. "..., and identified cold front using COAMPS data".

29. Line 372: the cyclone center is not visible.

We have changed line 372 to "Figure 9 shows the objectively identified cold front from the COAMPS…" so that it does not imply that the cyclone center is visible in Fig. 9.

30. Line 412: the location is too far north from the trades, the reference is therefore strange.

Our goal is to apply understanding (and compare properties) of the subtropical environment to midlatitude cyclone sectors, which is why we include this reference to the subtropics.

31. Lines 430-443: such high latent heat flux values are not uncommon in cold sectors of extratropical cyclones. Need to add relevant references on this.

As in the above comment, our intention is to distinguish the midlatitude cold sector region from other environments that produce boundary layer clouds (such as the subtropics), but we will also add context to our results in terms of other cold sectors.

"Li et al. (2021) find surface sensible and latent heat fluxes associated with CAO of 231 W m^-2 and 382 W m^-2, respectively, which is similar to those of this study."

32. The paragraph starting on line 445 and in the conclusion section attributes PBL height to the surface fluxes. However, the role of dry intrusions from the free troposphere were shown to enhance vertical mixing and PBL deepening by destabilizations, reconciling the strong inversion with enhanced vertical mixing and PBL height in the cold sector.

In cyclones with dry intrusions, the reason for the PBL deepening is also due to the enhanced surface fluxes. We see a deepening behavior here but do not know whether it is associated with a dry intrusion and enhancing the surface fluxes.

We have added a line in the paragraph on line 445 about the role dry intrusions may have on the cloud and boundary layer properties. "The strong subsidence in the cold sector helps to maintain the strong inversions. However, despite this strong subsidence, the boundary layer remains deep, suggesting that the PBL depth is maintained by the strong surface fluxes (Sinclair et al. 2010) and associated shallow convection, a process that could be enhanced by dry intrusions (Raveh-Rubin 2017)."

33. Line 498: at the end of section 5 there is a missing guidance of the reader: what can we learn about the relevance of the mechanisms outlined in the introduction from barotropic environments to the current case?

We agree and have added much more discussion of to what degree the subtropical environments are representative of the cold sector environment, namely the depth and level of decoupling of the boundary layer, as well as the role of surface fluxes. We reframe the paragraph beginning in line 519 of the conclusions so that it incorporates what we have learned from the subtropical environment applied to our case.

34. Line 542: Need to discuss the limitations of surface turbulent heat flux parameterizations.

The parameterization we are using is well-known to be a good measure of the surface fluxes and feel that an extensive discussion of surface parameterizations is outside the scope of the paper.

35. Figure 3: what are the green dashed lines? The field of 500-hPa height appear here with no context – need to add this field to Fig. 2.

The green dashed lines represent the bounds of the island influences (Ghate et al. 2021). This is stated in line 261, but we also add it to the figure caption.

36. Figure 4: This figure is out of context and can be moved to the supplementary material for smoother reading. Please also add a bar to show the variability among the different model runs. Note errors in the caption of the curve colors, and spell our MAE.

We actually think this is an important figure to show comparisons of the meteorology and analyze how COAMPS represents the spatial patterns of the synoptic system. This is a common and best method of showing the forecasting errors. COAMPS produces small errors in the large-scale meteorology and cyclone as a whole, suggesting that the model biases in the following paragraphs are not associated with errors related to the position or strength of the cyclone, and instead are due to smaller scale discrepancies or parameterizations. Though we have not removed the figure, we did describe its context better in lines 281.

"We quantify COAMPS forecast uncertainty by comparing 500-hPa geopotential height forecasts at various lead times against ERA5 reanalysis (Fig. 4) … The small error in meteorology suggests that the model biases discussed in the following paragraphs cannot be attributed to the error associated with the cyclone strength or phase."

We have also added error bars showing the standard deviation between model runs. We fixed the colors in the caption so that they match the colors in the plot and changed "MAE" to "mean absolute error".

The PBL height was determined by our hand-based best estimate for both COAMPS and soundings, but since this method isn't automated and therefore not practical to use throughout the rest of the analysis, we have changed the method for determining the PBL height depth (see response to Reviewer 2 SC4). We have also modified the tick marks on panel c so that they are the same on the x and y axes. In the caption, "MAE" has been changed to "mean absolute error".

Yes, some bins in the observation do not have data.

The observed fluxes are the ERA5 surface fluxes (line 149-150). PBL depth was from Richardson number to be consistent with observations but now is from the new method (see response to Reviewer 2 SC4). The change in PBL depth methods should clear up the confusion about which method is used in Fig. 13.

Though Fig. 12, 13, 14 shows the distributions of synoptic and cloud properties, Fig. 15 directly relates the synoptic properties to the clouds (LWP) in each sector, which is the entire goal of the paper. It also gives more insight into the environments in which COAMPS produces cloud.

Dry intrusions may be influencing the subsidence of the cold sector, but we have no way of identifying which points in the plot are influenced or not by dry intrusions. While the dry intrusion may be enhancing subsidence, surface fluxes, etc., separating the normal cold sector properties from the dry intrusion properties is not feasible without extensive back trajectory analysis, which is beyond the scope of this study.

However, we add the potential influence of dry intrusions: "Further behind the cold front, the EIS is larger, and the clouds are in an environment dominated by large-scale subsidence, reminiscent of a stratocumulus-topped boundary layer, and consistent with influences from dry intrusions."

Technical comments:
1. Line 49: the sentence is unclear, please rewrite.

The introduction has been rewritten, as well as this sentence.

2. Line 276: replace "all" with "and"

We have corrected this typo.

3. Line 332: remove minus sign

We have corrected km^-2 to km^2.

4. Figure 2: The cold front line is not visible. In caption – remove "t" from "start"

We have increased the color brightness of the cold front so that it is more visible and corrected the typo in the caption.

5. Figure 5 caption: remove "n" from "ration"

We have corrected the typo in the caption.

6. Figure 10: missing panel labels (a,b,…) referred to in the text.

We have added labels (a-f) to the subplots.

7. Figure 11 caption: what are the vertical lines?

The vertical lines represent the boundaries of the frontal sector. Per ACP guidelines, we have minimized the amount of descriptions in the caption that are clearly described in the figure. To us, the descriptions at the top of Figs. 11, 12 of the sector breakdowns are clear.

8. Line 466: change "clout" to "cloud"

We have corrected this typo.

9. Line 488: fix ".)" to ")."

We have corrected this typo.

References

Bretherton, C. S. and Wyant, M. C.: Moisture Transport, Lower-Tropospheric Stability, and Decoupling of Cloud-Topped Boundary Layers, Journal of the Atmospheric Sciences, 54, 148–167, https://doi.org/10.1175/1520-0469(1997)054<0148:MTLTSA>2.0.CO;2, 1997.

Galewsky, J.: Using stable isotopes in water vapor to diagnose relationships between lower-tropospheric stability, mixing, and low-cloud cover near the Island of Hawaii. Geophysical Research Letters, 45, 297–305. https://doi.org/10.1002/2017GL075770, 2018.

Ghate, V. P., Cadeddu, M. P., Zheng, X., and O'Connor, E.: Turbulence in The Marine Boundary Layer and Air Motions Below Stratocumulus Clouds at the ARM Eastern North Atlantic Site, Journal of Applied Meteorology and Climatology, https://doi.org/10.1175/JAMC-D-21-0087.1, 2021.

Hewson, T. D.: Objective fronts, Meteorological Applications, 5, 37–65, https://doi.org/10.1017/S1350482798000553, 1998.

Mass, C. F., D. Ovens, K. Westrick, and B. A. Colle: DOES INCREASING HORIZONTAL RESOLUTION PRODUCE MORE SKILLFUL FORECASTS?. *Bull. Amer. Meteor. Soc.*, 83, 407–430, https://doi.org/10.1175/1520-0477(2002)083<0407:DIHRPM>2.3.CO;2, 2002.

Naud, C. M., Booth, J. F., and Del Genio, A. D.: The Relationship between Boundary Layer Stability and Cloud Cover in the Post-Cold-Frontal Region, Journal of Climate, 29, 8129–8149, https://doi.org/10.1175/JCLI-D-15-0700.1, 2016.

Simmonds, I., Keay, K., and Tristram Bye, J. A.: Identification and Climatology of Southern Hemisphere Mobile Fronts in a Modern Reanalysis, Journal of Climate, 25, 1945–1962, https://doi.org/10.1175/JCLI-D-11-00100.1, 2012.

Uccellini, L. W., and D. R. Johnson: The Coupling of Upper and Lower Tropospheric Jet Streaks and Implications for the Development of Severe Convective Storms. Monthly Weather Review, 107, 682–703, https://doi.org/10.1175/1520-0493(1979)107<0682:TCOUAL>2.0.CO;2, 1997.

Wood, R. and Bretherton, C. S.: On the Relationship between Stratiform Low Cloud Cover and Lower-Tropospheric Stability, Journal of Climate, 19, 6425–6432, https://doi.org/10.1175/JCLI3988.1, 2006.

Zheng, X., Zhang, Y., Klein, S. A., Zhang, M., Zhang, Z., Deng, M., Tian, J., Terai, C. R., Geerts, B., Caldwell, P., and Bogenschutz, P. A.: Using Satellite and ARM Observations to Evaluate Cold Air Outbreak Cloud Transitions in E3SM Global Storm-Resolving Simulations, Geophysical Research Letters, 51, e2024GL109 175, https://doi.org/10.1029/2024GL109175, 2024.

---

## Referee Report (RR1)

**Comments on revisions**

The authors have gone through a lot of effort to address our comments, and the revisions have greatly improved the manuscript. The introduction now does a much better job at contextualizing the study within current research and defines the aims of the paper more clearly. By changing their two PBL-depth metrics to a single new metric, the authors have added clarity and avoided confusion, while also adding novelty to the study. The figures are more accessible and their descriptions more precise.

I found a sentence that I think might be incomplete (probably a word missing) in line 77: *"Therefore, regional models, which have much finer spatial and temporal resolution compared to observations, [MISSING WORDS?] to study PBL clouds in midlatitude cyclones and CAO."*

I recommend publishing the paper.

---

## Author Response (AR3)

We thank the reviewers and editor for the minor comments and catching the technical errors. We have addressed the comments below and corrected the manuscript.

**Anonymous referee #2 comments**

**I found a sentence that I think might be incomplete (probably a word missing) in line 77: "Therefore, regional models, which have much finer spatial and temporal resolution compared to observations, [MISSING WORDS?] to study PBL clouds in midlatitude cyclones and CAO."**
We have modified this sentence and added the missing words: "Therefore, we use regional models, which have much finer spatial and temporal resolution compared to observations, to study PBL clouds in midlatitude cyclones and CAO."

**Anonymous referee #4 comments**

**With all the change in the revised version the first aim of the paper as stated by the authors is still too broad for a paper that is based on a single case study (line105). I suggest setting it as the broader scope of the paper, and then setting the two aims to first evaluate the COAMPS model and second to examine the PBL and cloud properties across the front.**
We have modified the broader scope of the paper and the objectives as the reviewer suggested: "Here we apply the Naval Research Laboratory Coupled Ocean/Atmosphere Mesoscale Prediction System (COAMPS, Hodur 1997) regional model to a case study of a wintertime midlatitude cyclone over the Eastern North Atlantic. We apply our understanding of boundary-layer clouds, largely relevant to barotropic atmospheres in the subtropics, and expand it to boundary-layer clouds accompanying the midlatitude baroclinic synoptic system. We have two linked objectives: 1. to evaluate to what extent COAMPS represents the synoptic controls on cloud regime and PBL evolution within a midlatitude cyclone, including an analysis on the model's sensitivity to the microphysical parameterization; and 2. to examine the transitions of synoptic, cloud, and boundary layer properties across different regions of the cyclone in both observations and COAMPS, in a way that minimizes errors associated with cyclone strength and phase."

**Minor corrections -**
**Line 32 – missing the word "lower" in the sentence**
We have added "lower" before albedo in this sentence.
**Line 61 – the wording at the end of the sentence is unclear, please check**
We have modified this sentence to read: "They found that the post-cold-frontal region had stronger winds and subsidence compared to regions of northerly of northwesterly flow and subsidence that are far away and not associated with the cold front."
**Line 73 – missing "can be used"/"are used" before "...to study PBL clouds in midlatitude cyclones..."**

We have modified this sentence and added the missing words: "Therefore, we use regional models, which have much finer spatial and temporal resolution compared to observations, to study PBL clouds in midlatitude cyclones and CAO."

**Line 129 – remove one "combines"**

We have removed one of the "combines".

**Line 139 – missing year in the reference.**

We have added the year "2014" to the citation.

**Line 355-356 – The second part of the sentence is not clear (the reference?), rewrite**

We have modified this sentence to now read: "The reason for the larger COAMPS indices is because the coarse model resolution causes the top of the PBL to be above the inversion and therefore the upper PBL layer is warmer and drier than the surface layer."

**Figure 13 + 14 – the difference between the darker and lighter colors is not strong enough**

The dark colors representing COAMPS are already near saturation and the light colors representing the observations are as light as they can be without the colors being washed out. Any further lightening of them makes them too gray that their color cannot be seen. We have made many versions of this plot and tried many colors, and we believe this to be the best color scheme. Therefore we have not made any changes to Fig. 13 and 14

**Editor comments**

**In addition, I recommend to rework the abstract to sharpen the statements, in particular:**

**1) Please resolve the apparent contradiction in these two sentences: "However, the simulated clouds are mostly too thick, with too much liquid water and too little cloud-base drizzle,compared to observations from satellite and retrievals from ground-based observations at Graciosa Island in the Azores. The transects reveal a shallow, conditionally unstable boundary layer in the warm sector, accompanied by overly shallow clouds with low liquid water content."**

While the warm sector clouds in both the observations and model have lower liquid water content than clouds in the cold sector, the cold sector clouds in the model have much more liquid than the observed clouds. We have clarified that the underestimation of liquid water occurs in clouds in the warm sector while the overestimation of depth and liquid water occurs in the cold sector.

**2) "unique insights into cloud and boundary layer evolution throughout a cyclone." do you mean in the context of a cyclone, or throughout the life cycle of a cyclone?**

We mean in the context of the cyclone. In addressing comment #3 below, we have removed this sentence.

**3) Please add one sentence that clarifies the implications of your work for model evaluation and more generally for our understanding of the state and/or behaviour of the atmosphere and climate, which is the main requirement for publication in ACP.**

We have modified the final sentence to be more specific about the impacts of our study. "Our analysis framework serves as a unique approach to model verification, and our results offer

insights into differences in boundary-layer cloud behavior between subtropical and synoptic cold-sector regimes."

**Please remove one of the first of the two References section title and the empty page.**
We have removed the blank References page so now there is only one References header and no empty pages.